# Structural and functional investigations of *syn*-copalyl diphosphate synthase from *Oryza sativa*

Xiaoli Ma [1,5] ✉, Haifeng Xu[1,2,5], Yuru Tong[3], Yunfeng Luo[4], Qinghua Dong[1] & Tao Jiang [1,2] ✉

The large superfamily of labdane-related diterpenoids is defined by the cyclization of linear geranylgeranyl pyrophosphate (GGPP), catalyzed by copalyl diphosphate synthases (CPSs) to form the basic decalin core, the copalyl diphosphates (CPPs). Three stereochemically distinct CPPs have been found in plants, namely (+)-CPP, *ent*-CPP and *syn*-CPP. Here, we used X-ray crystallography and cryo-EM methods to describe different oligomeric structures of a *syn*-copalyl diphosphate synthase from *Oryza sativa* (OsCyc1), and provided a cryo-EM structure of OsCyc1[D367A] mutant in complex with the substrate GGPP. Further analysis showed that tetramers are the dominant form of OsCyc1 in solution and are not necessary for enzyme activity in vitro. Through rational design, we identified an OsCyc1 mutant that can generate *ent*-CPP in addition to *syn*-CPP. Our work provides a structural and mechanistic basis for comparing different CPSs and paves the way for further enzyme design to obtain diterpene derivatives with specific chirality.

[1] National Laboratory of Biomacromolecules, Institute of Biophysics, Chinese Academy of Sciences, Beijing, China. [2] University of Chinese Academy of Sciences, Beijing, China. [3] School of Pharmaceutical Sciences, Capital Medical University, Beijing, China. [4] School of Traditional Chinese Medicine, Capital Medical University, Beijing, China. [5] These authors contributed equally: Xiaoli Ma, Haifeng Xu. ✉email: maxiaoli@ibp.ac.cn; tjiang@ibp.ac.cn

Terpenes, or terpenoids, are the most abundant natural products on Earth and are widely used as energy, fuel materials and drugs such as artemisinin and triptolide[1–3]. Terpenes can be classified by the number of carbons as monoterpenes (C10), sesquiterpenes (C15), diterpenes (C20) and triterpenes (C30). Diterpenes are stereochemically and structurally diverse compounds that comprise nearly 10,000 known species in which the 20-carbon isoprene units fold into different backbones. Diterpene synthases can be categorized into two classes, with class I diterpene synthases being ion-initiated, while class II diterpene synthases are proton-initiated. Although not all diterpene biosynthesis processes involve class II diterpene synthases, most medicinal plants produce diterpenes with a labdane skeleton, and their synthesis involves the catalysis of class II diterpene synthases[4]. Copalyl diphosphates (CPPs) are labdane backbone diterpenes that are produced by class II diterpene synthases (diTPSs), the copalyl diphosphate synthases (CPSs), by catalyzing the general precursor (E, E, E)-geranylgeranyl pyrophosphate (GGPP)[5]. In plants, the most common stereoisomers of CPPs are ent-CPP and (+)-CPP, while syn-CPP is less commonly observed (Fig. 1a)[5,6]. In the common cereal rice (Oryza sativa), syn-copalyl diphosphate synthase (also commonly known as OsCPS4 or OsCyc1) catalyzes GGPP to syn-CPP[7–9] (Fig. 1a). Many of the phytoalexins and allelopathic agents in rice, such as momilactone A/B and oryzalexin S, come from syn-CPP and represent the only known metabolic fates for this compound[7,10–15].

The catalytic mechanisms of terpene synthases share several common characteristics. Catalytic reactions of terpene synthases are multistep cascade reactions with multiple carbocation intermediates and carbocation rearrangements occur through a 1,2-hydride shift and/or alkyl shift to form new skeleton backbone products[16]. Class II terpene synthases have the characteristic DXDD motif in which the middle aspartic acid generally acts as the catalytic acid that initiates the reaction by protonation, and

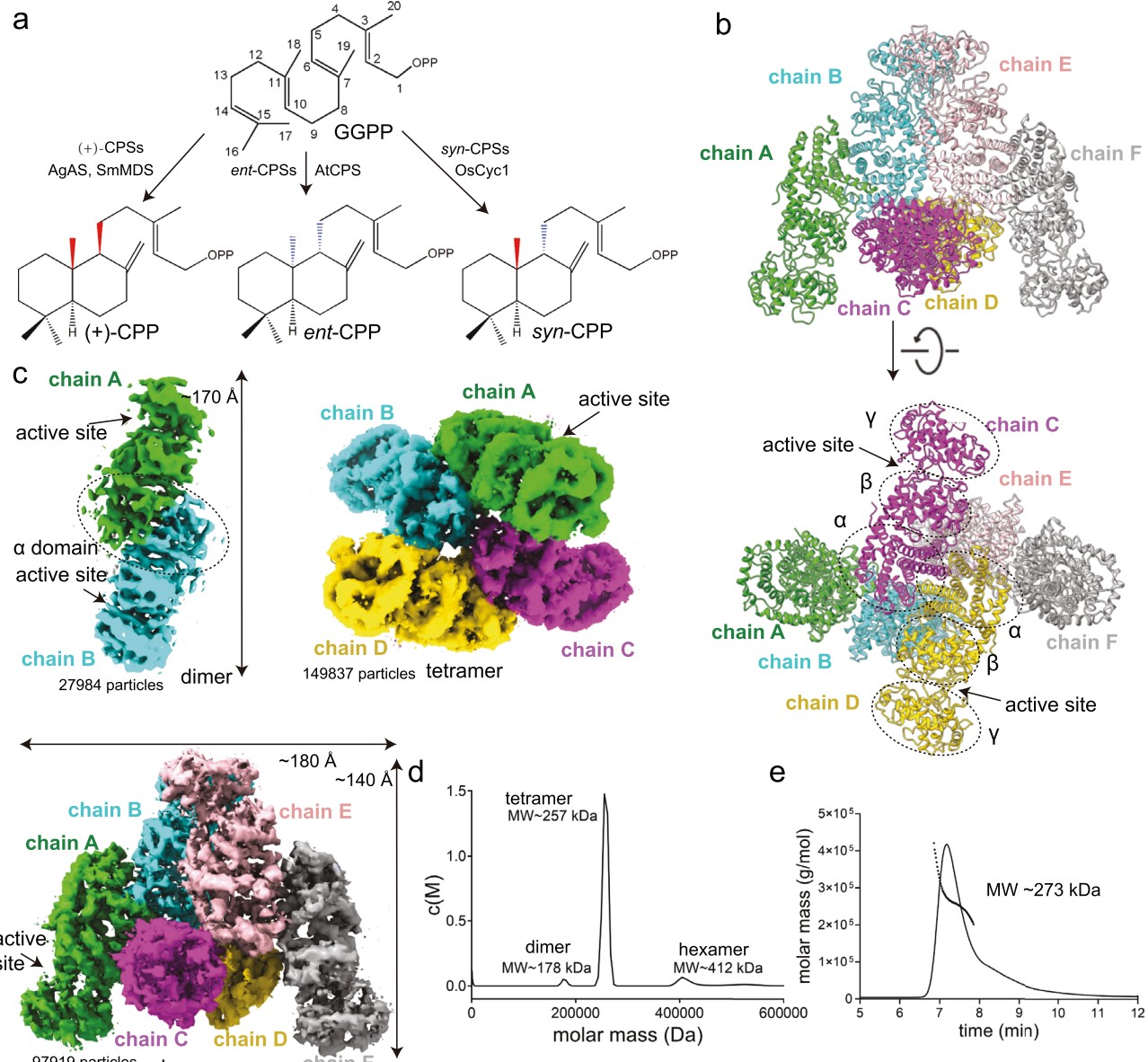

**Fig. 1 Structure overview and properties of OsCyc1. a** GGPP is catalyzed by various CPSs to produce *syn*-CPP, (+)-CPP and *ent*-CPP. **b** Crystal structure of OsCyc1. The α, β and γ domains are indicated by separate circles. Chains A, B, C, D, E and F are colored lime green, cyan, magenta, yellow, pink and dark gray, respectively. **c** Cryo-EM maps of the dimer, tetramer and hexamer of OsCyc1. **d** Analytical ultracentrifugation analysis of OsCyc1[69-767]. **e** Profile of the static light scattering experiment of OsCyc1.

the catalytic base is variable[17,18]. The mechanisms underlying class II diTPSs resemble other terpene synthase reaction mechanisms, but also have their own characteristics. In plants, three stereoisomers of CPPs are all obtained via the intermediate labda-13$E$-en-8-yl$^+$, whose stereochemistry depends on the *pro-chiral* conformation of the substrate GGPP[5]. However, due to the absence of *syn*-CPS structure, the specific structural and mechanistic differences in the diterpene synthases that produce the three stereochemically distinct CPPs remain to be elucidated. For example, while class II diTPSs usually deprotonate directly, they also terminate by capturing a water molecule[19,20]. Two studies investigating the histidine-asparagine dyad in *ent*-CPP producing diterpene synthase (*ent*-CPS) and tyrosine-histidine dyad in (+)-CPP producing diterpene synthase ((+)-CPS) show that mutating histidine to alanine produces hydroxylated derivatives of CPPs[21–23]. Another study, however, found that changing the corresponding histidine (H251) to alanine in *syn*-CPS OsCyc1 did not alter its product to hydroxylated CPP[18]. Due to the lack of structural information for *syn*-CPSs, there is no structural explanation for why OsCyc1-related amino acid mutations do not produce hydroxylated CPPs. In addition, the common or individual mechanisms of different stereochemical CPPs producing CPSs remain elusive.

The crystal structure of plant *ent*-CPS from *Arabidopsis thaliana* (AtCPS) complexed with a substrate analog revealed the substrate binding mode for *ent*-CPS[24]. The apo state structure of abietadiene synthase from *Abies grandis* (AgAs)[25] and substrate binding mode structure of miltiradiene synthase from *Selaginella moellendorffii* (SmMDS)[26] showed the structures of (+)-CPSs. However, the structure of plant *syn*-CPS remains absent. Here, we set out to uncover the structure of OsCyc1 and explore structural and mechanistic differences in CPSs that produce stereochemically different CPPs using structural biology, mass spectrometry, molecular docking, and bioinformatics methods. In summary, we solved the crystal structure of wild-type OsCyc1, together with three cryo-EM structures of OsCyc1 in different oligomeric forms and a cryo-EM structure of the OsCyc1$^{D367A}$ mutant in complex with the substrate GGPP. Importantly, we obtained a mutant that produces *ent*-CPP in addition to the primary product *syn*-CPP.

## Results

**X-ray and Cryo-EM structures of OsCyc1**. To determine the structure of OsCyc1, we used the stable N-terminal truncation variant of OsCyc1, OsCyc1$^{69–767}$, from which the plastid targeting peptide was removed. This resulted in a pseudo-mature protein that was catalytically active as shown by our relative enzyme activity assay (Supplementary Figs. 1 and 2). Next, we determined the crystal structure of OsCyc1$^{69–767}$ by molecular replacement method at 3.5 Å resolution, which contains six protomers (Chains A/B/C/D/E/F) in the asymmetric unit (Fig. 1b, Supplementary Table 1 and Supplementary Data 1). When we used size exclusion chromatography, we found that both full-length OsCyc1 and truncated OsCyc1$^{69–767}$ exhibited similar peak positions near 11 mL on a Superdex 200 HR 10/300 column, indicating that OsCyc1 and OsCyc1$^{69–767}$ both assemble as oligomers in solution (Supplementary Fig. 1b, c). To elucidate the oligomer states and the interaction mode of OsCyc1, we determined the cryo-EM structure of OsCyc1 and finally solved three OsCyc1 structures in dimer, tetramer and hexamer forms, at a resolution of 7.9 Å, 3.5 Å and 3.7 Å, respectively (Fig. 1c, Supplementary Fig. 3 and Supplementary Table 2). Furthermore, we have also determined the structure of OsCyc1$^{D367A}$ in its hexameric form with GGPP, at a resolution of 4 Å (Supplementary Fig. 3c, d and Supplementary Table 2).

The overall structure of monomer OsCyc1 is similar to that of AtCPS, AgAS and SmMDS, with RMSD values for Cα of 1.86 Å, 2.15 Å and 1.94 Å, respectively (Supplementary Fig. 4a). The structure of monomer OsCyc1 consists of an αβγ domain and a catalytic center located between the β and γ domains (Fig. 1b). The catalytic centers are located at the outside edge of the dimer, tetramer and hexamer cryo-EM structures (Fig. 1c). The dimer is composed of protomer A and protomer B, and the dimerization interface involves their α domains (Fig. 1c). The tetramer is composed of two dimers that consist of chains A/B and chains C/D (Fig. 1c). Our analysis showed that the cryo-EM structure of hexamer is almost identical to the structure observed in the asymmetric unit of the X-ray crystal structure of OsCyc1. The six molecules in OsCyc1 hexamer structures were labeled as chains A, B, C, D, E and F, which can be divided into three dimers. Two of these dimers assemble like two sides of an isosceles triangle (molecule A/B and molecule E/F), and the other dimer (molecule C/D) lies in the center of the isosceles triangle (Fig. 1c). The tetramer molecules correspond to subunits A/B/C/D in the hexamer.

The molecular weights of monomeric OsCyc1 and OsCyc1$^{69–767}$ are 88.2 kDa and 80.8 kDa, respectively. When we performed analytical ultracentrifugation analysis, we found that OsCyc1$^{69–767}$ exhibits various oligomerization states, and the main molecular weight is between those of the trimer and tetramer (Fig. 1d). Static light scattering experiments showed that the calculated molecular weight of OsCyc1 is ~ 273 kDa (Fig. 1e), which is also a value between that of the trimer and tetramer. In the particle picking process of Cryo-EM structure determination, tetrameric assemblies are the major particles that are picked. Combined with the cryo-EM structure results, these findings imply that the major oligomeric form of OsCyc1 in solution (as used in our experiments) is tetrameric.

**Biochemical analysis of the OsCyc1 oligomer state**. To identify the key residues responsible for oligomer stability, and to assess the effects of oligomerization on OsCyc1 activity in vitro, we investigated the biochemical properties of full-length wild-type OsCyc1 and its mutants. Initially, we assessed the relative enzyme activity of OsCyc1 at different concentrations of Mg$^{2+}$. According to reports, Mg$^{2+}$ is essential for class I terpene synthases, while for class II terpene synthases, an excessive amount of Mg$^{2+}$ is generally not required, and trace amounts of Mg$^{2+}$ are necessary[27,28]. Moreover, it was reported that in class II sesquiterpene cyclase, magnesium ions function as co-factors, typically two with every substrate[29]. In our experiments, in the absence of additional magnesium ions in the solution, OsCyc1 demonstrates its highest level of enzyme activity. The presence of extra magnesium ions partially inhibits the enzyme's activity (Supplementary Figs 2 and 4b). However, when EDTA is used to chelate potential magnesium ions, the activity of OsCyc1 is significantly inhibited (Supplementary Fig. 4c, e). In addition, we assessed the influence of temperature on enzyme activity, and the optimal temperature of 16 °C was selected for determining relative enzyme activity of OsCyc1 and its mutants was estimated by the peak area calculated from GC-MS experiments.

As the crystal structure of OsCyc1 was solved at a higher resolution than its cryo-EM structures, particularly in the γ domain region, we focused on utilizing the crystal structure to analyze oligomer interactions. Additionally, since the cryo-EM structure of OsCyc1$^{D367A}$ revealed clear densities of amino acid side chains, we also presented the density details of cryo-EM structure of OsCyc1$^{D367A}$ to illustrate the differences between the crystal and cryo-EM structures. As shown in Fig. 2a, the

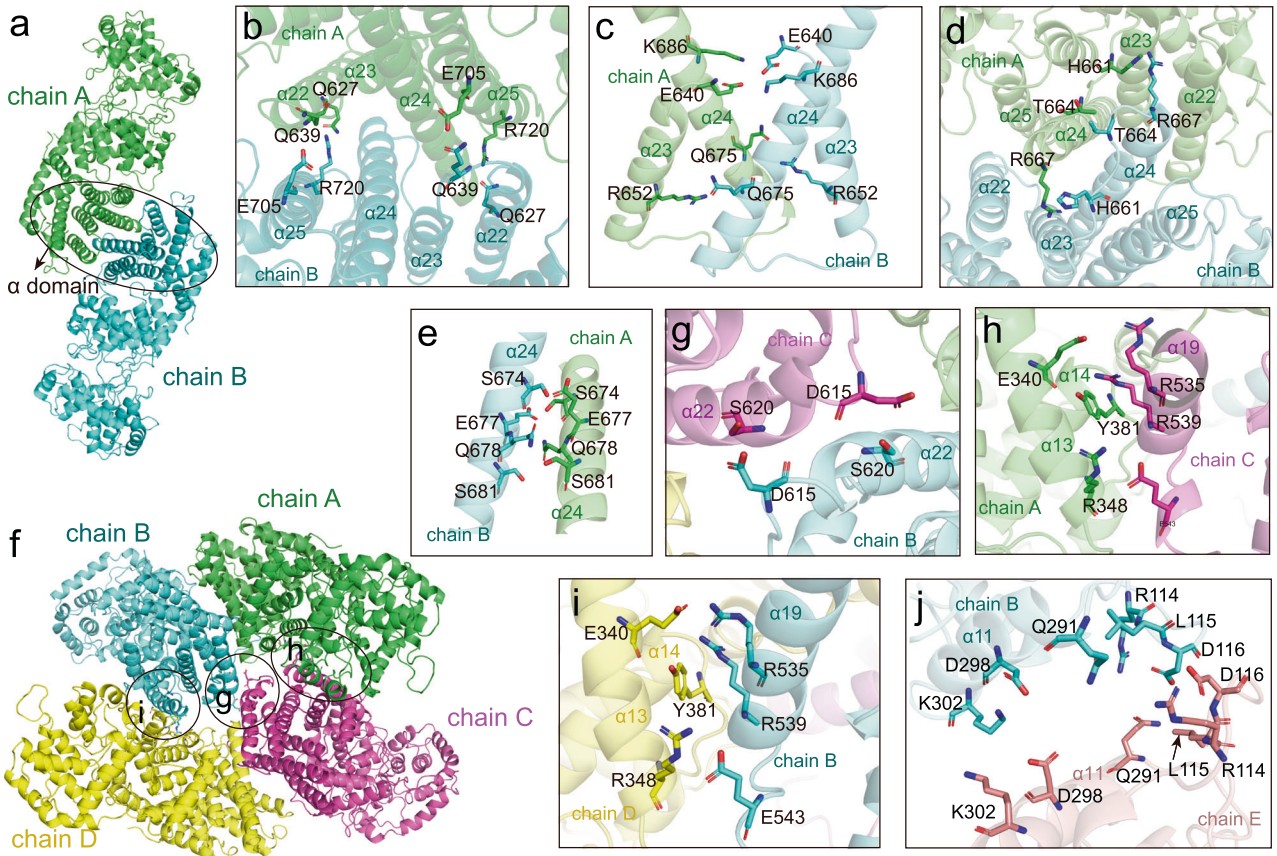

**Fig. 2 Detailed views of the OsCyc1 oligomer interface. a–e** Detail of the dimer interface, which shows the α domain interactions of chain A and chain B. **f–i** Views of tetramer interface showing residues involved in protomer interactions. **j** Interactions between chain B and chain E in the hexamer structure.

interactions in the dimer assembly of protomers/chains A and B occur symmetrically in α domain helices α22, α23, α24 and α25 (Fig. 2a, b). Residues located within or near α22 interact with residues in α25, such as Q627 with R720, which are at a hydrogen bonding distance in the crystal structure (Fig. 2b and Supplementary Fig. 5a, b). However, the side chain densities for Q639 and E705 are absent, making it challenging to accurately predict their interaction forces (Fig. 2b and Supplementary Fig. 5c, d). The interactions between α23 and α24 include R652 with Q675, as well as E640 with K686, possibly through van der Waals interactions (Fig. 2c and Supplementary Fig. 6a-c). Furthermore, the residues H661 and R667, located in the loop between α23 and α24, may also contribute to dimer formation (Fig. 2d and Supplementary Figs. 6d, 7a). In addition, we found that H-bonds form between S674 and E677 within the α24 helix (Fig. 2e and Supplementary Fig. 7b). Furthermore, there may be van der Waals interactions between Q678 and S681 in the α24 helix (Fig. 2e and Supplementary Fig. 7c).

When we analyzed the particle numbers in the cryo-EM map reconstruction, we found that the tetramer is the predominant form of OsCyc1 and is equivalent to chains A/B/C/D in the crystal structure (Fig. 1c and Supplementary Table 3). Apart from the dimer interface, intermolecular interactions also occur between chain B and chain C, chain A and chain C, and chain B and chain D (Fig. 2f). Interactions between chain B and C are mainly located on helix α22 and involve residues S620 and D615 (Fig. 2g and Supplementary Fig. 7a, d). In chain A, β domain residues in or near α13 and α14, such as E340, R348 and Y381, interact with α domain residues of chain C in or near α19, such as R535, R539 and E543, which are identical to the interactions between chain B and chain D (Fig. 2h, i and Supplementary

Fig. 8). When we analyzed the binding interactions in the crystal structure, we found that the interactions between different dimers are similar. The interaction between dimer C/D and dimer E/F is identical to the interaction between dimer A/B and dimer C/D, suggesting that a pseudo-twofold symmetry axis exists between the chains C and D (Supplementary Fig. 4d). In addition to the abovementioned interactions in the center of the hexamer, a linkage exists in the γ domain of protomer B and protomer E, which protrude to the edge of the crystal hexamer structure. The interactions between chain B and chain E occur on α11 and the loop between α2 and α3, including key residues D298 and K302 (Fig. 2j and Supplementary Fig. 9a). R114, D116, and Q291 might also participate in the interactions between chain B and chain E. However, due to the poor densities in these regions, it is challenging to analyze their interactions (Fig. 2j and Supplementary Fig. 9b–d).

To investigate the effects of oligomerization on enzyme activity, we selected key amino acids for mutation experiments to disrupt the binding state of OsCyc1 in solution. Most alanine substitution mutations showed size exclusion chromatograph profiles similar to that of the WT, albeit with lower relative enzyme activity (Fig. 3a and Supplementary Fig. 10). Q291 is located at the chain B-E interface, and its substitution by alanine showed the lowest relative enzyme activity compared to others. A static light scattering analysis of the Q291A protein, purified from the size exclusion chromatograph, showed that most of the Q291A proteins present a ~660.6 kDa state (Fig. 3b). This observation suggests that mutations may destabilize the primary tetramer form in solution and cause a higher degree of oligomerization state. The S674A/E677A mutant form exhibited the highest relative enzyme activity, comparable to that of WT,

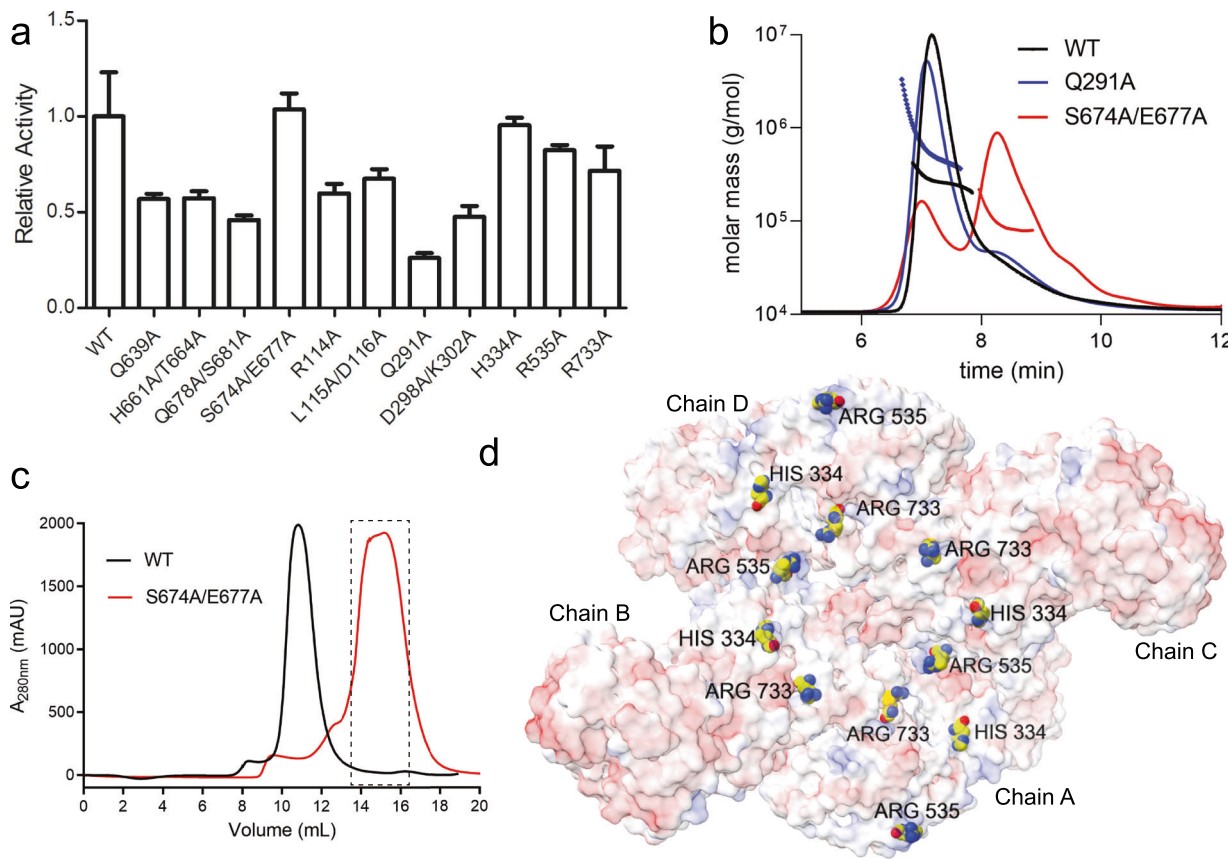

**Fig. 3 Analysis of oligomeric properties of OsCyc1. a** Relative enzyme activity of amino acid mutants at the interface of OsCyc1 oligomers. The error bar represents the mean with SEM. **b** Profile of the static light scattering experiment of WT, Q291A and S674A/E677A. **c** Size exclusion chromatography of WT (OsCyc1$^{69-767}$) and S674A/E677A mutant. Protein samples were loaded onto a Superdex 200 HR 10/300 column for gel filtration assays. **d** Electro potential map of the tetramer. H334, R535 and R733 are shown as spheres in the cartoon representation. Red is for negative potential, white at zero, and blue is for positive in the electro potential map.

and showed a lagging peak position in the size exclusion chromatograph (Fig. 3c), indicating that S674A/E677A is present in a state of lower oligomerization. The S674A/E677A protein from the peak marked by the dotted box in Fig. 3c was collected for the static light scattering experiment, which showed a major ~112.4 kDa peak (Fig. 3b). Residues S674 and E677 are both located in the original dimer interface and form hydrogen bonds (Fig. 2e and Supplementary Fig. 7b). Considering the results of our size exclusion chromatography and the static light scattering experiments, we hypothesized that S674A/E677A disrupts the oligomeric state of wild-type OsCyc1. Together, these results imply that the tetramer is not essential for enzyme activity in vitro.

Next, we investigated whether oligomerization affects local substrate or product concentrations near enzymes through the interaction of diterpene pyrophosphate with surface positive amino acid residues. Therefore, we selected surface-exposed positive residues, such as H334, R535 and R733, located in the center of the structure for mutagenesis studies (Fig. 3a, d). However, the replacement of these residues with alanine showed only a slight effect on enzyme activity (Fig. 3a and Supplementary Fig. 11), which is in agreement with the result observed in the S674A/E677A mutant, showing that the tetramer seems to have a minor impact on enzyme activity in vitro.

**Structure of GGPP in *Syn*-Copalyl Diphosphate Synthase OsCyc1.** The DXDD motif is located in the active site of class II terpene synthases. Mutation of the middle aspartic acid to

alanine has been found to result in inactivation of enzyme activity. To investigate the substrate binding mode, GGPP was incubated with OsCyc1$^{D367A}$ before cryo-EM sample preparation. We solved the cryo-EM structure of the OsCyc1$^{D367A}$ mutant in complex with substrate GGPP at a resolution of 4 Å (Fig. 4a and Supplementary Fig. 3c, d and 12a). As exemplified by hexamer protomer B, GGPP harbors a continuous and clear density in the active site (Fig. 4b). We found that while the overall architecture of OsCyc1$^{D367A}$ is similar to that of wild-type OsCyc1, a number of differences exist around the active site, mainly in the loop near residue N405 and helix α7 (Fig. 4a and Supplementary Fig. 12a).

Next, we compared the ligand binding state structures of (+)-CPS SmMDS (PDB ID: 4Y47)[26], *ent*-CPS AtCPS (PDB ID: 3PYA)[24] and OsCyc1$^{D367A}$. We found that SmMDS possesses the largest active pocket mainly due to structure elements corresponding to helix α7 and the loop near residue K453 in OsCyc1, while the pocket size of OsCyc1$^{D367A}$ is the smallest and has more structural features similar to AtCPS (Fig. 4c, Supplementary Figs 12b and 13). When we analyzed the structure of AtCPS, we found that the phosphate part of substrate analog AG8 in the AtCPS structure is more flexible and protrudes to the outside of the pocket. In the OsCyc1$^{D367A}$ structure, the α7 helix and loops near residues D192 and N405 embrace GGPP more compactly (Fig. 4d and Supplementary Fig. 12c). Moreover, the inner pocket electrostatic potential of AtCPS is more negative than that of OsCyc1 (Supplementary Fig. 13).

In the structure of OsCyc1, K233, N405 and K453 interacts with the pyrophosphate, while M194 and E199 are close to the

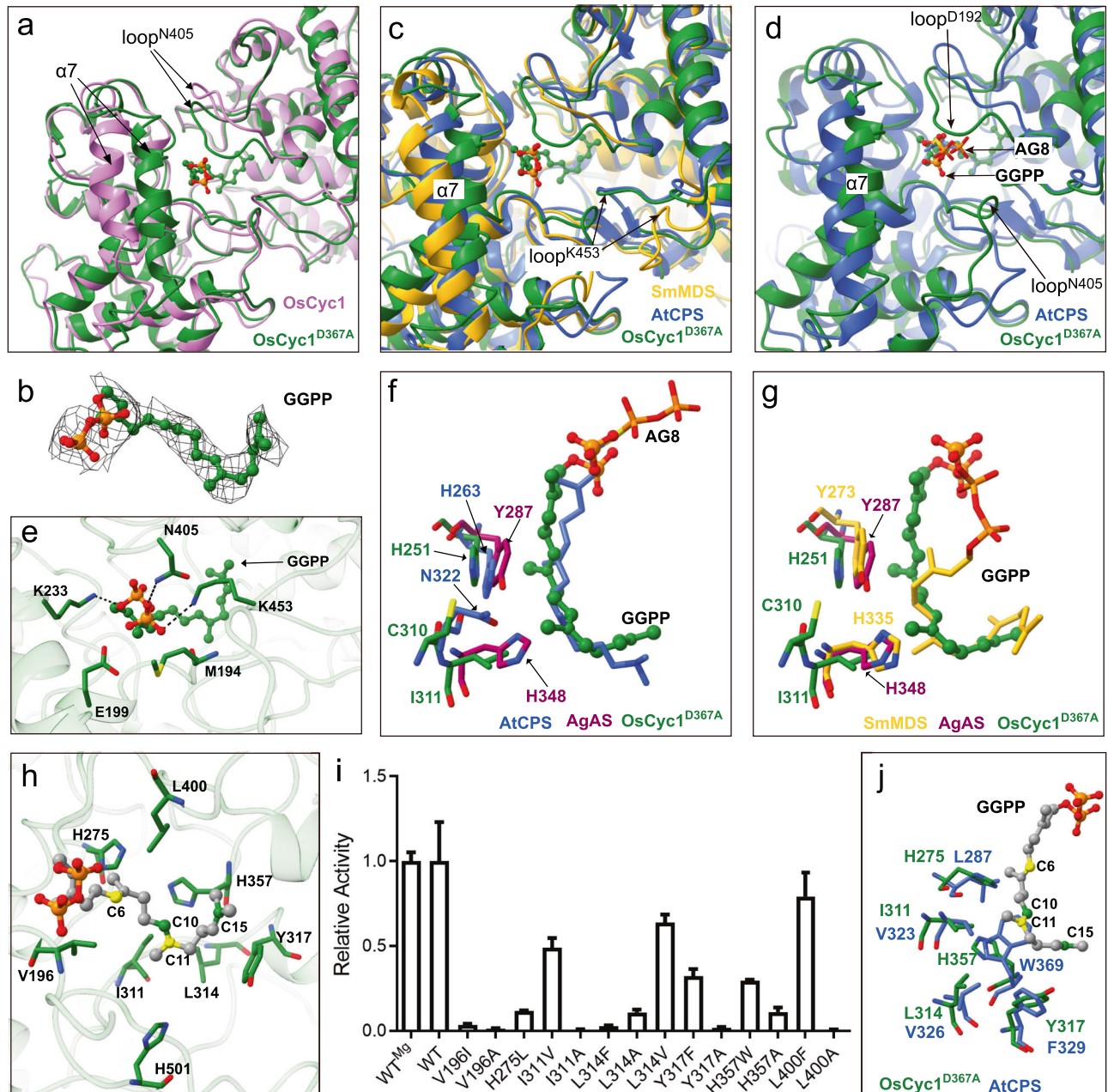

**Fig. 4 Structure comparison of OsCyc1 and related proteins. a** Active pocket comparison of OsCyc1 (orchid) and OsCyc1[D367A] (forest green). **b** Cryo-EM density of GGPP in the chain B of OsCyc1[D367A] structure. The density is shown as a contour level of 0.0192 and a step of 1. **c** Comparison of the structures in their ligand binding states of SmMDS (goldenrod, PDB ID: 4Y47), OsCyc1[D367A] (forest green) and AtCPS (royal blue, PDB ID: 3PYA). **d** Comparison of active site pocket structures of OsCyc1[D367A] (forest green) and AtCPS (royal blue, PDB ID: 3PYA). **e** Key residues in the substrate binding pocket of OsCyc1[D367A] that contribute to pyrophosphate binding. **f, g** Position of the catalytic dyad in AgAS, AtCPS, SmMDS and OsCyc1[D367A]. **h** Key residues around GGPP in OsCyc1[D367A]. C6 and C11 of GGPP are shown in yellow. C10 and C15 of GGPP are shown in forest green. **i** Relative enzyme activity of active pocket amino acid mutants. The error bar represents the mean with SEM. **j** Comparison of active site pocket residues between OsCyc1[D367A] (forest green) and AtCPS (royal blue, PDB ID: 3PYA). In the structure of AtCPS, AG8's pyrophosphate exhibits two conformations. For image clarity, we have opted to display only one conformation.

pyrophosphate of GGPP (Fig. 4e and Supplementary Fig. 14a, b). In terpene synthases, the middle aspartic acid of the DXDD motif is typically stabilized by nearby basic amino acids. For instance, in squalene-hopene cyclase (SHC) from *Alicyclobacillus acidocaldarius*[30] and bacterial diterpene synthases such as PtmT2[31], the basic amino acid is histidine, whereas in plant class II terpene synthases, it is asparagine[32,33] (Supplementary Fig. 15). In OsCyc1[D367A] structure, the DXDD motif and the conserved basic amino acid, asparagine (N414), are situated at the

terminus of the isoprene tail of GGPP. The distance between the d2-oxygen atom in D365 and the carbon-20 atom in the isoprene tail of GGPP is 2.59 Å (Supplementary Fig. 14d). However, measuring the distance between the unmutated D367 and the substrate is not feasible due to its mutation to alanine. In the crystal structure of OsCyc1, the distance between the d1-oxygen in D367 and the d2-nitrogen in the conserved N414 is 2.97 Å (Supplementary Fig. 14e, f). Upon superimposing the crystal structure and the OsCyc1[D367A] structure, the measured distance

between the d2-oxygen in D367A and the carbon-19 in GGPP is 2.82 Å, which is suitable for initiating the cyclization reaction (Supplementary Fig. 14g). However, in the OsCyc1$^{D367A}$ structure, GGPP adopts a linear conformation, possibly reflecting the stage of binding, which might not represent the cyclization state.

The stereochemistry of copalyl diphosphate constitutes the main reason for the diversity observed among diterpene structures. Previous studies have shown that mutating key residues of terpene synthase either alters the reaction pathway, or alternatively, generates new products[21,34–36]. In previous reports, a histidine-asparagine dyad (H263, N322) in the ent-CPS AtCPS and a tyrosine-histidine dyad (Y287, H348) in (+)-CPS AgAS were identified as the catalytic base dyad. In both cases, mutating histidine to alanine produced hydroxylated CPP[21–23] (Fig. 4f and Supplementary Fig. 15). In comparison, another study found that mutating the possible catalytic dyad (H251A, H251D and C310A) in OsCyc1 did not change the product to hydroxylated CPP[18]. Together, these findings suggested that OsCyc1 harbors different active pocket property compared to other stereochemically distinct CPPs producing CPSs. To assess whether OsCyc1 possesses a specific amino acid sequence in the active pocket, we aligned sequences of class II diterpene synthases that produce CPPs and hydroxylated CPPs (8-hydroxycopalyl diphosphate and peregrinol diphosphate; Supplementary Figs 15, 16). However, it was not possible to distinguish product determinants between different types of enzymes. This observation is in line with a previous study that showed that it is difficult to interpret functional discrepancies based only on sequence analysis of terpene synthases[37].

Among the active pockets of AtCPS, AgAS, SmMDS and OsCyc1, that of OsCyc1 constitutes the smallest pocket. In OsCyc1, the side chains of H251, C310 and I311 are positioned at distances of 3.7 Å, 6.9 Å and 3.7 Å, respectively, from GGPP. In AtCPS, the corresponding amino acids have distances to AG8 of 2.6 Å, 4.6 Å and 4.2 Å, respectively, while in SmMDS, these distances are 3.5 Å, 6.3 Å and 3.6 Å, respectively (Supplementary Table 4). Therefore, compared to the structures of AtCPS and SmMDS, the residues H251, C310 and I311 of OsCyc1 are located relatively distant from the substrate (Fig. 4f, g). One study found that deprotonation is more easily achieved than hydroxylation, except that water is efficiently replenished and must attack close to the axis of the carbon cation empty 2p orbital in triterpene synthesis[20]. Terpene synthases share many similarities in their reaction mechanisms. On the basis of these findings, we hypothesize that the small size of the active pocket and the long distance between the possible catalytic dyad and the substrate may together contribute to the inability of OsCyc1 to produce hydroxylated CPP by mutation of the corresponding residues.

We next analyzed the structural determinants that may be related to product chirality. Given that the size of the active pocket and substrate position are similar of OsCyc1 and AtCPS, we assumed that it is more likely to change product syn-CPP to ent-CPP than (+)-CPP through enzyme design. When we analyzed the structure of OsCyc1$^{D367A}$, we found that the isoprene tail of GGPP is surrounded by negative and hydrophobic residues (Fig. 4h and Supplementary Fig. 13). Through sequence and structure alignment with (+)-CPSs and ent-CPSs, the specific residues of OsCyc1 that may be involved in catalytic reactions were identified as V196, H275, I311, L314, Y317, H357, L400 and H501(Fig. 4h, Supplementary Figs 14c-k and 15). In previous report, substitution of H501 in OsCyc1 has been well studied showing that H501A, H501D and H501F produce syn-halima-5,13E-dienyl diphosphate in an E.coli modular metabolic engineering system[18] (Supplementary Fig. 15c). Our results showed that while the relative enzyme activity of V196A,

I311A, Y317A and L400A mutants were nearly eliminated, the mutants L314A and H357A retained partial enzyme activity (Fig. 4i, Supplementary Figs 11 and 17 and Supplementary Table 5). Mutating L314 to a hydrophobic residue containing a longer side chain such as phenylalanine, also impaired enzyme activity. In contrast, mutations of residues to the corresponding residues in AtCPS, namely, H275L, I311V, Y317F, H357W and L400F, resulted in high relative enzyme activity, except for V196I, which exhibited little relative enzyme activity (Fig. 4i and Supplementary Figs 11 and 17). While we investigated all the different amino acid mutants that around the substrate, none of the abovementioned single site mutants produced additional new products that could be detected in our experiments.

**Product Stereochemistry Change Through Site-Directed Mutagenesis.** To explore the possibility of structural-based product chirality alteration, we next performed multiple-site mutation experiments. OsCyc1 catalyzes GGPP to form syn-CPP, in which new bonds form between C6 and C11, as well as between C10 and C15 (Fig. 1a and Fig. 4h, j). By analyzing OsCyc1$^{D367A}$ structure, we observed that H275, I311, L314, Y317, H357 and L400 were close to the reaction carbon atoms (Fig. 4h). In addition, mutating these residues to corresponding residues in AtCPS only slightly impaired enzyme activity (Fig. 4i). On the basis of these findings, we next focused on these residues to investigate the effects of multiple-site mutations on the conformation of OsCyc1 product.

As shown in Fig. 4h, H275, H357 and Y317 line up along C6 to C11 of GGPP. The dual site mutants H275L/H357W, Y317F/H357W and H275L/Y317F did not produce detectable products, while the triple site mutant H275L/Y317F/H357W produce trace amounts of syn-CPP (Supplementary Table 6 and Supplementary Fig. 18). In addition, H275L/H357W/L400F and H275L/Y317F/H357W/L400F produce syn-CPP, while H275L/I311V/Y317F or H275L/C310D/I311V/Y317F did not produce detectable products (Supplementary Table 6 and Supplementary Fig. 18). These findings suggest that if the intermediate amino acid H357 is excluded, the combination of amino acids H275 and Y317 cannot catalyze the production of syn-CPP efficiently.

I311, L314, Y317 and H357 are at the isoprene tail of GGPP in OsCyc1$^{D367A}$ structure. The mutation combination of I311, L314, Y317 and H357, such as L314V/Y317F/H357W, I311V/Y317F/H357W and I311V/L314V/Y317F/H357W produce trace amounts of syn-CPP or no detectable product (Supplementary Table 6 and Supplementary Fig. 18). The addition of L400F mutation did not improve the relative enzyme activity for the above mutants (Supplementary Table 6 and Supplementary Fig. 18). These results indicate that the bottom of active pocket plays an important role in enzyme activity.

Interestingly, a five-site mutant (H275L/I311V/L314V/Y317F/H357W, namely, OsCyc1$^{5Mu}$) and a six-site mutant (H275L/I311V/L314V/Y317F/H357W/L400F), produce syn-CPP (peak 1) as well as a new product (peak 2) (Fig. 5a, e, f and Supplementary Fig. 18). Through analyzing the mass spectrum of the GC-MS 275 m/z chromatograph peak, we found that peak 1 and peak 2 share similar mass spectra (Fig. 5e, f). As previously reported, the dephosphorylated derivatives of (+)-CPP and ent-CPP exhibited identical retention times, different from the retention time of the dephosphorylated derivative of syn-CPP[8]. In addition, the GC-MS 275 m/z chromatograph peaks of the dephosphorylated derivatives of the three CPP stereoisomers all had the same mass spectrum[7,8]. By comparing peak 2 with authentic standards of ent-CPP and (+)-CPP, we inferred that peak 2 corresponds to either (+)-CPP or ent-CPP (Fig. 5a–f). To distinguish between

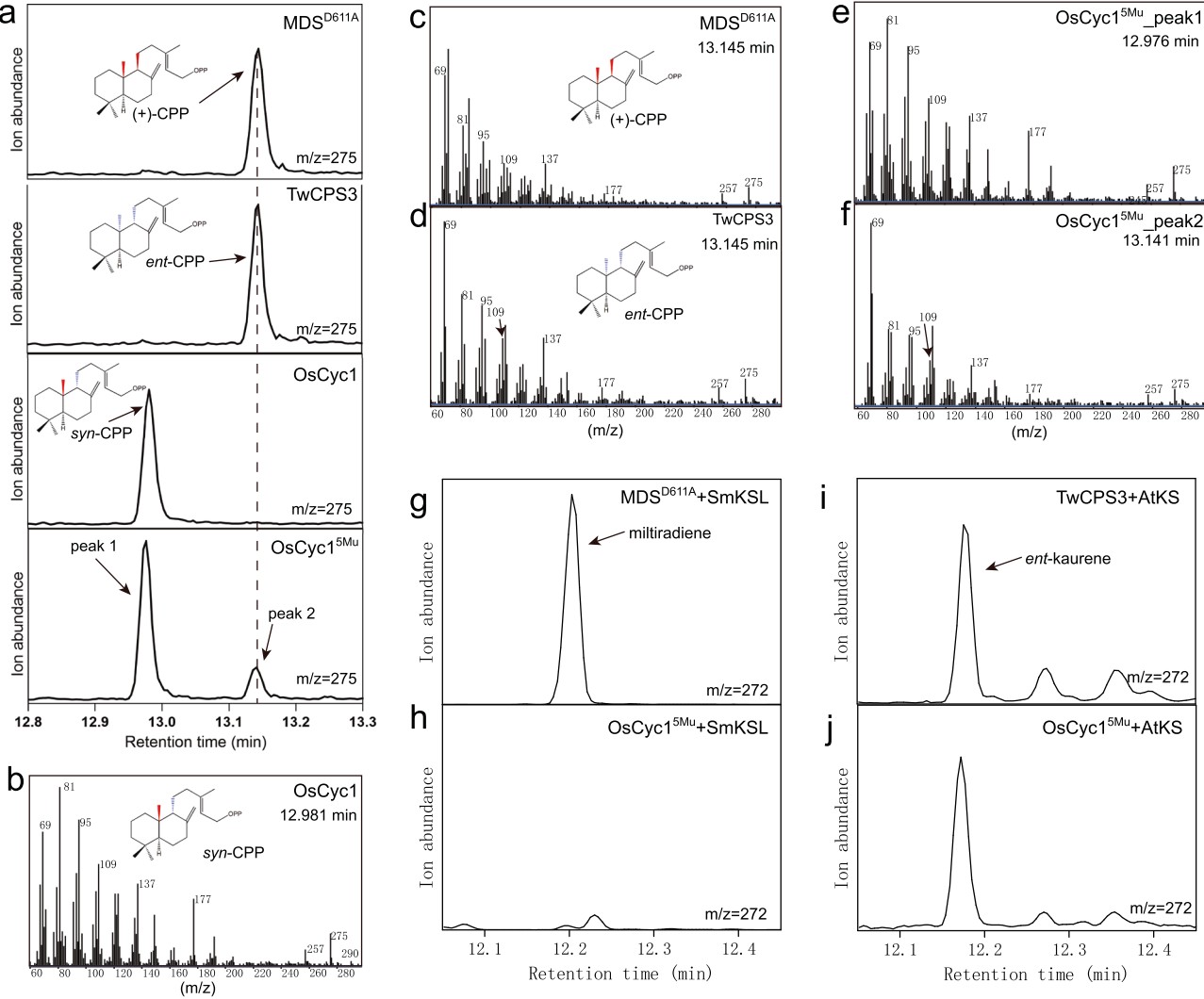

**Fig. 5 GC-MS analysis of enzyme reaction products. a** Extracted ion chromatograms (m/z = 275) from GC-MS analysis of the dephosphorylated reaction products from MDS[D611A], TwCPS3, OsCyc1 and OsCyc1[5Mu]. **b, c, d, e, f** Mass spectrum of the GC-MS 275 *m/z* chromatograph peak. The background mass spectrometry was subtracted from the mass spectrometry. **g, h, i, j** GC-MS analysis (272 *m/z* extracted ion chromatographs) of the products from MDS[D611A] and SmKSL, OsCyc1[5Mu] and SmKSL, TwCPS3 and AtKS, and OsCyc1[5Mu] and AtKS.

these two possibilities, we performed enzyme catalysis experiments using OsCyc1[5Mu] together with AtKS or SmKSL. As shown in Fig. 5g and i, AtKS catalyzes *ent*-CPP to *ent*-kaurene[38,39] and SmKSL catalyzes (+)-CPP to miltiradiene[40,41], which were used as authentic standards. Finally, by comparing the retention time and mass spectrum of the catalysis product, we confirmed that the new product corresponding to peak 2 is *ent*-CPP. (Fig. 5g–j and Supplementary Fig. 19).

In our experiments, we found that multi-site mutants containing only I311V, L314V, Y317F and H357W had no detectable enzyme activity, while mutants with an additional H275L produced *syn*-CPP. Furthermore, OsCyc1[5Mu] can generate not only *syn*-CPP but also *ent*-CPP. To better understand how amino acids affect the stereochemistry of the product, we docked *syn*-CPP, (+)-CPP and *ent*-CPP into the structure of OsCyc1[5Mu] (See Method). Our docking results showed that the new bond-forming atoms (C6, C10, C11 and C15) of *syn*-CPP, *ent*-CPP and (+)-CPP were located near I311, L314, Y317 and H357 (Fig. 6a–c). The side chain of I311 was found to cause steric hindrance with C16 of both (+)-CPP and *ent*-CPP, preventing the formation of these two stereochemically distinct CPPs. H275, although distant from the decalin ring of CPPs, was found to be

close to the C6 atom of GGPP in the OsCyc1[D367A] structure (Supplementary Fig. 20a). In addition, the phosphate positions of (+)-CPP and *ent*-CPP were in conflict with the side chain of H275, while the phosphate of *syn*-CPP adopted a different position (Fig. 6a–c). The negative effects of the mutations containing I311V, L314V, Y317F and H357W on the enzyme activity can be alleviated by H275L, possibly by reshaping the substrate binding pocket shape to accommodate the linear region of the product, thus facilitating the reaction. These results indicated that H275 plays an important role in *syn*-CPP production. This speculation is in line with the comprehensive bioinformatics analysis of *syn*-CPSs, *ent*-CPSs and (+)-CPSs in plants, which shows that H275 is relatively conserved in *syn*-CPSs and different from other CPSs (Fig. 6e and Supplementary Figs 21–23).

Furthermore, we performed docking of *syn*-CPP into the OsCyc1[D367A] structure (Fig. 6d and Supplementary Fig. 20b). Based on the results of structural analysis and molecular docking, we proposed a structural mechanism model of the enzyme catalysis reaction, in which key residues around the nearest atom of GGPP or *syn*-CPP are indicated (Fig. 6f). The active site key amino acids, namely H275, I311, L314, Y317, and H357, play a

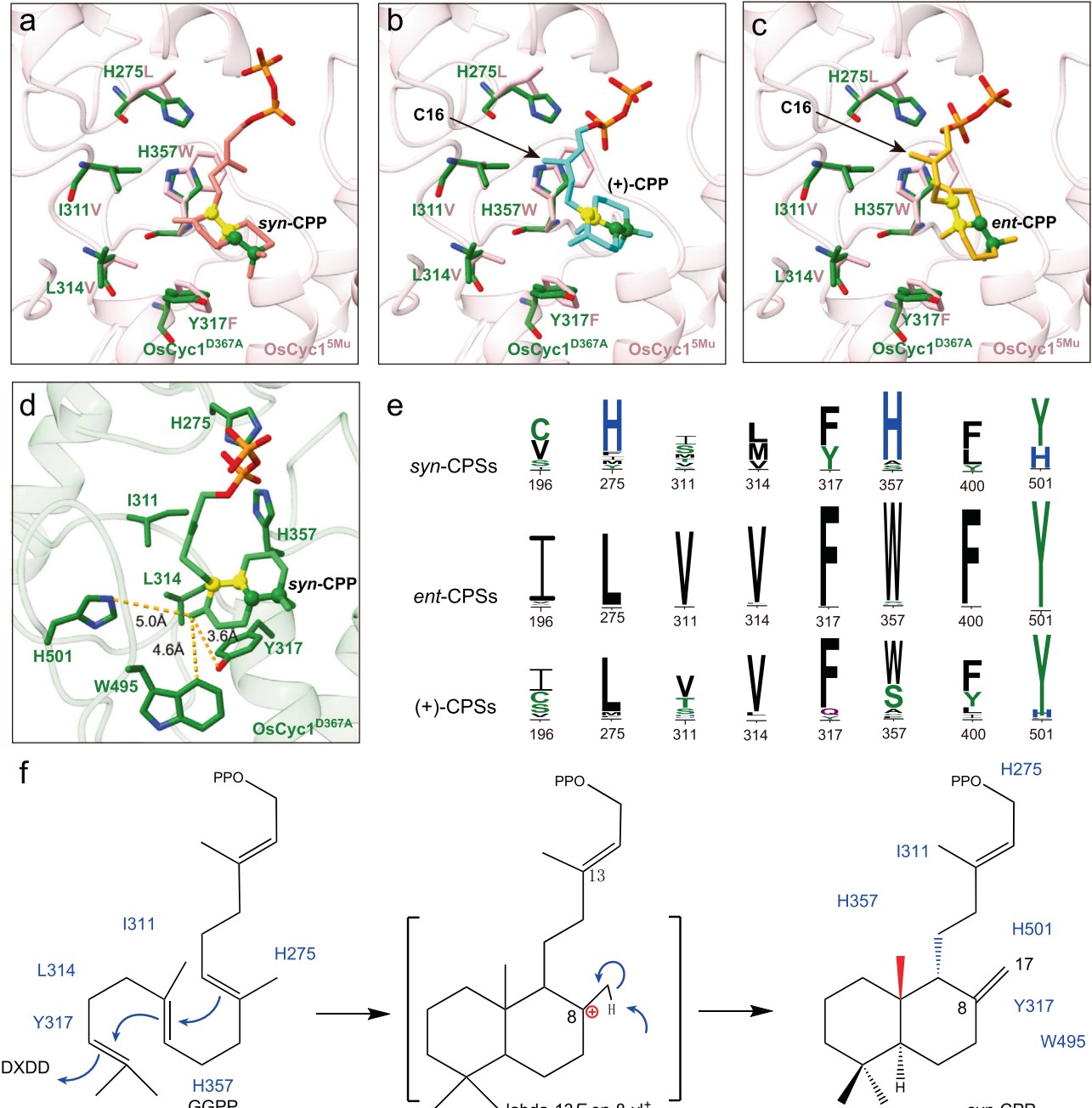

**Fig. 6 Molecular docking and bioinformatics information of OsCyc1. a, b, c** The molecular docking results of *syn*-CPP (salmon), (+)-CPP (cyan) and *ent*-CPP (goldenrod) to the active site of OsCyc1$^{5Mu}$. Structures of OsCyc1$^{D367A}$ (forest green) and OsCyc1$^{5Mu}$ (light pink) are aligned. **d** The molecular docking results of *syn*-CPP (lime green) to the active site of OsCyc1$^{D367A}$. **e** Sequence logos highlighting the conservation of V196, H275, I311, L314, Y317, H357, L400 and H501 in *syn*-copalyl diphosphate synthases, *ent*-copalyl diphosphate synthases and (+)-copalyl diphosphate synthases. Each logo consists of stacks of symbols, one stack for each position in the sequence. The height of symbols within the stack indicates the relative frequency and information content in bits. Sequence numbers are labeled according to OsCyc1 protein sequence. **f** The cyclization mechanism of GGPP to form *syn*-CPP. Amino acids near GGPP or *syn*-CPP are labeled according to the structure or molecular docking results.

crucial role in determining the stereochemical conformation of *syn*-CPP. Among these, I311, L314, Y317 and H357, positioned in the proximity of the decalin ring of *syn*-CPP, are responsible for folding of GGPP into the specific conformation of *syn*-CPP. Specially, I311 causes steric hindrance to prevent the formation of other stereochemical CPPs. Additionally, H275, which is highly conserved in *syn*-CPSs, restricts the conformation of *syn*-CPP by regulating the position of pyrophosphate. In conclusion, H275, I311, L314, Y317 and H357 jointly regulate the conformation of *syn*-CPP.

Interestingly, in the molecular docking results of *syn*-CPP and OsCyc1$^{D367A}$, Y317, W495, and H501 are close to the C17 atom of *syn*-CPP (Fig. 6d). These amino acids may be involved in the carbon cation quenching of the intermediate lambda-13*E*-en-8-yl$^{+}$ (Fig. 6d, f). Previous report showed that mutations of H501 to aspartic acid or phenylalanine can generate *syn*-halima-5,13*E*-dienyl diphosphate[18]. *Syn*-halima-5,13*E*-dienyl diphosphate is the product quenched from the intermediate halima-13*E*-en-10-yl$^{+}$, which is derived from the rearrangement via 1,2-hydrid and methyl shift of the intermediate lambda-13*E*-en-8-yl$^{+42}$

(Supplementary Fig. 20c). Based on these findings, we speculated that H501 may be the final base residue that extracts the proton from the intermediate lambda-13$E$-en-8-yl$^+$.

## Discussion

Here, we report the *syn*-copalyl diphosphate synthase structure and show different oligomer state structures of αβγ domain terpene synthases. Previous studies indicated that terpenoid synthases composed of the αβ domain are always identified as monomers or dimers in solution[43], and terpenoid synthases composed of only the α domain can also assemble as tetramers or hexamers in solution[43,44]. OsCyc1 is the αβγ domain CPS with a 3D structure determined in tetramer and hexamer forms, which provides a good template for a detailed investigation of its assembly mechanism and functional implications. It has been predicted that successive steps in a metabolic pathway often colocalize and multidomain terpenoid synthases assemble into oligomers to facilitate catalysis through cluster channeling[43,45] or electrostatic channeling[39] mechanisms. Therefore, we performed a number of mutation studies to assess these two possibilities in vitro. Unfortunately, our study failed to obtain direct evidence supporting these two theories for OsCyc1. However, the in vivo situation could be more complicated. Considering that OsCyc1 is a monofunctional diterpene synthase that catalyzes the reaction in its only active pocket, we infer that oligomerization may facilitate its interaction with its interacting partner proteins in vivo, and thus enhance catalysis efficiency. This speculation needs to be verified by in vivo studies.

In this study, we reported the GGPP-bound structure of OsCyc1$^{D367A}$. The active pocket shows marked conformational changes compared with the *apo*-state structure of OsCyc1. Compared to CPSs that produce other stereochemically different CPPs, namely, AtCPS, SmMDS and AgAS, the possible catalytic base dyad of OsCyc1 is distant from the substrate, and the substrate binding pocket size of OsCyc1 is compact, which potentially facilitates the folding of the substrate GGPP into a less stable chair-boat conformation[5]. Together, these two structural properties provide a reasonable explanation for the unchanged product type of possible catalytic base dyad mutagenesis (H251A, H251D or C310A) experiments in OsCyc1[18] and emphasize the importance of terpene structure-based mechanism studies.

Previous studies showed that the product stereochemistry of class II diterpene cyclization depends on the *pro*-chiral conformation of GGPP[5]. The five-site mutant OsCyc1$^{5Mu}$ mimics key residues in AtCPS and can produce both *syn*-CPP and *ent*-CPP. Subsequently, the docking results and bioinformatic analysis indicated H275 and I311 play an important role in CPP stereochemistry determination, while H501 may serve as the base residue accepting the proton from the intermediate lambda-13$E$-en-8-yl$^+$. Although we obtained another stereochemical CPP through enzyme design, the lower enzyme activity inhibits the further application of this mutant. Our results emphasize that in addition to the rational design of products, ensuring enzyme activity is an important but challenging aspect.

In conclusion, the structural and mechanistic research of OsCyc1 provides detailed insights into the active site pocket for *syn*-copalyl diphosphate synthase and sheds light on the pivotal differences in structural elements for *syn*-CPP and other conformation CPPs producing CPSs. Our results clearly demonstrated that the copalyl diphosphate chirality can be changed only by the simultaneous substitution of more than 3 or 4 residues, supporting the theory that the fate of GGPP is determined by the shape of the active pocket[43]. These results could be useful for future enzyme design to obtain CPP derivatives with desired chirality.

## Methods

**Gene cloning, protein expression and purification**. Codon-optimized genes encoding the full-length OsCyc1 protein and truncated protein (OsCyc1$^{69-767}$) were synthesized (GENEray biotechnology) and subcloned into pET-24 a (+) vectors between the *NdeI* and *XhoI* sites, including a His-tag. These constructs were subsequently transformed into BL21(DE3) cells and grown in LB medium at 37 °C until the optical density (OD)$_{600}$ reached 0.6-1.0. Protein expression was induced by addition of 0.2 mM isopropyl β-D-thiogalactoside (IPTG) for 18h-20h at 16 °C. Full-length OsCyc1 and its mutants were purified by Ni affinity chromatography followed by Superdex$^{TM}$ 200 HR 10/300 gel filtration. OsCyc1$^{69-767}$ and its mutant OsCyc1$^{D367A}$ were purified by Ni affinity chromatography followed by Resource Q chromatography and Superdex$^{TM}$ 200 HR 10/300 gel filtration. All proteins were stored in buffer A (20 mM Tris-HCl pH 8.0, 200 mM NaCl, 2 mM DTT), except for H275L, I311V, Y317F, H357W, L400F, WT and OsCyc1$^{D367A}$ proteins used for structure studies or relative enzyme activity assays, which were in buffer B (20 mM Tris-HCl pH 8.0, 200 mM NaCl, 2 mM DTT, 5 mM MgCl$_2$). Protein expression levels were assessed by SDS-PAGE and quantified by Nanodrop.

The D611A mutant of partially truncated miltiradiene synthase from *Selaginella moellendorffii* (SmMDS$^{D611A}$) and truncated kaurene synthase like from *Salvia miltiorrhiza* (SmKSL) were cloned into the pET-24 a (+) vectors between the *NdeI* and *XhoI* sites, including a His-tag. The construct was subsequently transformed into BL21(DE3) cells and grown in LB medium at 37 °C until the optical density (OD)$_{600}$ reached 0.6-1.0. Protein expression was induced by addition of 0.2 mM isopropyl β-D-thiogalactoside (IPTG) for 18h-20h at 16 °C. SmMDS$^{D611A}$ was purified by Ni affinity chromatography.

TwCPS3 from *Tripterygium wilfordii* (obtained from Gao's laboratory) was cloned into the pET-22b vector between the *NdeI* and *XhoI* restriction sites, including a His-tag. The construct was subsequently transformed into BL21(DE3) cells and grown in LB medium at 37 °C until the optical density (OD)$_{600}$ reached 0.6-1.0. Protein expression was induced by addition of 0.2 mM isopropyl β-D-thiogalactoside (IPTG) for 18h-20h at 16 °C. The recombinant protein was purified by Ni affinity chromatography and stored in buffer B.

The gene encoding *ent*-kaurene synthase from *Arabidopsis thaliana* (AtKS) was codon-optimized and synthesized (from TsingkeBiotechnologyCo., Ltd.) into pET-24 a (+) vector between the *NdeI* and *XhoI* sites. Subsequently, the constructed vector was transformed into BL21(DE3) cells. To induce expression, isopropyl β-D-thiogalactoside (IPTG) was used, and the recombinant protein was subsequently purified by Ni affinity chromatography. Finally, the recombinant protein was stored in buffer B.

**Site-directed mutagenesis**. The mutants were constructed by PCR using an overlap extension strategy with full-length OsCyc1 pET-24 a (+) vector as a template. After PCR, *Dpn*I was used to digest the wild-type template for 1 hour at 37 °C. The digested product was then transformed into *E. coli* DH5α chemically competent cells and selected mutants were verified by sequencing.

**Crystallization, data collection and structural determination**. OsCyc1$^{69-767}$ was concentrated to 10 mg/mL and crystalized in reservoir solution (2% v/v 1,4-dioxane, 0.1 M Tris pH 8.0, 15% w/v polyethylene glycol 3,350) by the hanging drop method at 18 °C using equal volumes of protein and crystallization reservoir solution. The crystals were cryoprotected in reservoir solution with 20% glycerol and flash-frozen in liquid nitrogen before data

collection under cryogenic conditions (100 K) at Shanghai Synchrotron Radiation Facility (SSRF) beamline BL18U1.

Diffraction data were indexed, integrated and scaled in XDS[46]. The crystal belongs to space group $P2_12_12_1$ with a hexamer in the asymmetric unit. The structure was solved by molecular replacement using the structure of AtCPS (PDB ID: 3PYA, the model was modified by Chainsaw program in the CCP4 program suite[47]) as a searching model by Phaser-MR (full-featured) in PHENIX[48] with the number of copies set to 4. Subsequently, Phaser-MR provided an interpretable electron density map, and the model was built using Phenix.Autobuild in PHENIX. The resulting structure was iteratively manually modeled and modified using Coot[49] and refined by Phenix.Refine in PHENIX[48]. The refinement strategy employed XYZ (reciprocal-space) and XYZ (real-space) refinement methods, along with Individual B-factors and Group B-factors. Additionally, non-crystallographic symmetry (NCS) was used. The strategy changed as the model improved. The statistics were listed in Supplementary Table 1. Structures were prepared in PyMOL (http://www.pymol.org) or UCSF ChimeraX[50].

**Cryo-EM sample preparation and data collection**. (*E, E, E*)-Geranylgeranyl pyrophosphate (GGPP, Sigma Aldrich, St. Louis, Missouri, USA) at a final concentration of 0.1 mM was soaked with the OsCyc1[D367A] mutant before cryo-EM sample preparation. The proteins of OsCyc1 and OsCyc1[D367A] were concentrated to 1.8 mg/mL and 3 mg/mL in buffer B for cryo-EM sample preparation, separately. 3 μL sample was applied to R 1.2/1.3 Au 300 mesh holey carbon films (Quantifoil Micro Tools) after pretreated with glow discharge. The grids were blotted for 3 s at force 2 in a 4 °C, 100% humidity chamber and frozen *via* plunge-freezing using liquid ethane by Vitrobot Mark IV (Thermo Fisher Scientific). The frozen grids were stored in liquid nitrogen for further analysis.

Data collection was performed using a 300 kV FEI Titan Krios microscope equipped with Gatan K3 direct electron detector. Images were recorded by beam-image shift data collection methods[51]. The camera was in super-resolution mode and the physical pixel size is 1.07 Å (22500X). For WT OsCyc1, 3287 micrographs were recorded with 60 e/Å² total dose. For OsCyc1[D367A], 3939 micrographs were recorded with 60 e/Å² total dose.

**Cryo-EM data processing, model building and validation**. The images were motion corrected and dose-weighted by MotionCor2[52]. Image processing was calculated within RELION-3.1.1[53] and CTF estimation was performed with Gctf[54]. Automatic particle picking was performed using the Laplacian-of-Gaussian and picked particles were subjected to a reference-free 2D classification. Subsequently, good particles were selected as 2D references for particle picking. For WT OsCyc1, after several rounds of 2D classification, dimers, tetramers and hexamers were selected separately and subjected for initial model. After 3D classification and 3D auto-refine, post-processing was used. Finally, 27984 particles were refined to 7.9 Å for dimer and 97919 particles were refined to 3.7 Å for hexamer, both of which were C1 symmetry. For tetramer refinement, 149837 particles and C2 symmetry were used, which was refined to 3.5 Å.

For OsCyc1[D367A] structure, the data were processed with a similar procedure except hexamer is used as the initial model for 3D classification. OsCyc1[D367A] used 735747 particles and generated a 4.0 Å resolution map.

The crystal structure of OsCyc1 was used as initial model and manually inspected and rigid-body fitted into the maps in Chimera[55]. Then the models were adjusted in Coot. The 3D

conformer of GGPP was downloaded from Grade Web Server (http://grade.globalphasing.org/cgi-bin/grade/server.cgi) and real-space refined in Coot. Models were refined to maps using PHENIX[48] and validated using PHENIX. The statistics were listed in Supplementary Table 2.

**Analytical ultracentrifugation and static light scattering analysis**. Analytical ultracentrifugation was performed on a Beckman's ProteomeLab XL-I at 25 °C using OsCyc1[69–767] protein at an OD 280 nm of 0.8 in buffer B (20 mM Tris-HCl pH 8.0, 200 mM NaCl, 2 mM DTT, 5 mM MgCl₂). Data were collected at 40,000 r.p.m. every 3 min at a wavelength of 280 nm and the interference sedimentation coefficient distributions, c(M), were calculated from the sedimentation velocity data using SEDFIT[56].

Static light scattering (SLS) analysis was performed on a DAWN HELEOS II instrument (Wyatt Technology, Santa Barbara, CA) using a Nanofilm SEC-150, 4.6X300mm column from Sepax Technologies. The full-length OsCyc1 protein was diluted to 1.0 mg/mL in buffer A (20 mM Tris-HCl pH 8.0, 200 mM NaCl, 2 mM DTT). The light scattering detector was calibrated with BSA monomer standard before the assays and data were analyzed with ASTRA software (Wyatt Technology).

**Production of authentic standards of *ent*-CPP and (+)-CPP**. Studies found that D611A mutant of miltiradiene synthase from *Selaginella moellendorffii* (MDS[D611A]) produces (+)-CPP[57] and TwCPS3 catalyzes GGPP to *ent*-CPP[58]. On the basis of these findings, we used (+)-CPP produced by MDS[D611A] and *ent*-CPP produced by TwCPS3 as authentic standards.

**Relative enzyme activity assays**. OsCyc1 mutants used in relative enzyme activity assay were full-length and purified by Ni affinity chromatography followed by Superdex™200 HR 10/300 gel filtration except for proteins that were used in new product identification experiments. OsCyc1 proteins were collected in buffer A (20 mM Tris-HCl pH 8.0, 200 mM NaCl, 2 mM DTT), except H275L, I311V, Y317F, H357W and L400F which were collected in buffer B (20 mM Tris-HCl pH 8.0, 200 mM NaCl, 2 mM DTT, 5 mM MgCl₂). The relative enzyme activity experiments were performed in a 100 μL solution A, except for H275L, I311V, Y317F, H357W and L400F which were performed in solution B.

For the determination of the relative enzyme activity of the mutant variants used for enzyme activity comparison, solutions were supplemented with 2.5 μg GGPP and 50 μg protein, and the reactions were allowed to proceed for 2 h at 16 °C. For the determination of the relative enzyme activity of H275L/H357W/L400F, solutions were supplemented with 10 μg GGPP and 300 μg protein, and the reactions were allowed to proceed for 2 h at 16 °C. For the determination of the relative enzyme activity of mutants in Supplementary Fig. 18a–c and 18e (except for OsCyc1), solutions were supplemented with 30 μg GGPP and 600 μg protein, and the reactions were allowed to proceed for 2 h at 16 °C.

Two units of alkaline phosphatase (calf intestinal (CIP) (NEB)) were used for dephosphorylation for 2 h. Then the reaction solution was extracted with 600 μL n-hexane twice. The n-hexane extract was evaporated to dryness under a gentle nitrogen flow and then resuspended by 80 μL n-hexane.

For the determination of relative enzymatic activity of OsCyc1 under varying concentrations of magnesium ions, 2.5 μg GGPP and 40 μg of protein were incubated for 2 hours at 16 °C. Different magnesium ion concentrations were achieved by adding additionally 0 mM (control group), 2 mM, 5 mM, 10 mM, 20 mM, 30 mM, and 50 mM magnesium chloride to buffer A. For both the control group and reactions with added magnesium

ions, each reaction was performed in triplicate. Alternatively, the enzymatic activity was assessed with 5 μg GGPP and 100 μg of protein in the presence of buffer A with the additional supplementation of 1 mM, 5 mM, and 10 mM ethylenediamine-tetraacetic acid tetrasodium salt (EDTA). Reactions involving the addition of EDTA were performed once for each reaction.

The identification of new product of OsCyc1$^{5Mu}$ was carried out in buffer B, containing 30 μg GGPP and 300 μg OsCyc1$^{5Mu}$, at 16 °C overnight. After being dephosphorylated by alkaline phosphatase, the product was extracted three times, with each extraction using 900 μL n-hexane.

200 μg MDS$^{D611A}$ and 200 μg SmKSL were utilized to catalyze 10 μg GGPP, and the resulting product was used as a standard for miltiradiene. Similarly, 300 μg TwCPS3 and 200 μg AtKS were used to catalyze 10 μg GGPP, and the product was served as a standard for ent-kaurene.

For the identification of the new product of OsCyc1$^{5Mu}$, two separate reactions were conducted. In the first reaction, 600 μg OsCyc1$^{5Mu}$ and 600 μg SmKSL were used to catalyze 100 μg GGPP in buffer B overnight, with a total volume of 500 μL. In the second reaction, 600 μg OsCyc1$^{5Mu}$ and 600 μg AtKS were used to catalyze 100 μg GGPP under the same conditions. After each reaction, the product was extracted three times, with each extraction using 900 μL of n-hexane.

In the above experiments, the n-hexane extract was evaporated to dryness under a gentle nitrogen flow and then resuspended in 80 μL of n-hexane to obtain the test sample. GC-MS analysis was carried out on a DB-5MS column using helium as a carrier gas. 5 μL samples were injected. Initial oven temperature was set at 50 °C for 2 min followed by a 20 °C /min gradient to 300 °C and held for 10 min.

**Computational modeling**. The structures of (+)-CPP, ent-CPP and syn-CPP were downloaded from the PubChem website (https://pubchem.ncbi.nlm.nih.gov) in .sdf formats and then converted to .pdb formats in PyMOL. All molecules were prepared to .pdbqt formats for docking in AutoDockTools 1.5.6[59]. CPPs, OsCyc1$^{D367A}$ and OsCyc1$^{5Mu}$ were docked using AutoDock Vina[60]. OsCyc1$^{5Mu}$ was generated by mutating corresponding residues (H275L, I311V, L314V, Y317F, H357W and A367D) of OsCyc1$^{D367A}$ in Coot[49], in which the structures of OsCyc1 and AtCPS (PDB ID: 3PYA) were used as references.

**Bioinformatics**. Sequences used in the conservation analysis of 8-hydroxycopalyl diphosphate synthases were obtained from the Uniprot website using relevant keywords such as copal-8-ol diphosphate hydratase, 8-hydroxycopalyl diphosphate synthase, and 8-LPP. Then, the sequences were selected based on their record names, sequence lengths greater than 470 amino acids, and domains containing β and γ domains. Additionally, CfTPS2 (UniProtKB: X4ZWN5, diterpene synthase TPS2) and AbCAS (UniProtKB: H8ZM73, bifunctional cis-abienol synthase) were also added to the sequences for alignment. The sequence conservation analysis was performed using the WebLogo3 website.

Sequence alignments of syn-copalyl diphosphate synthases (16 protein sequences), ent-copalyl diphosphate synthases (94 protein sequences) and (+)-copalyl diphosphate synthases (62 protein sequences) for sequence similar network (SSN) analysis were achieved using the Enzyme Function Initiative (EFI) Enzyme Similarity Tool (EST)[61] with OsCyc1 sequence as the query sequence, taxonomy filter of "Eukaryota, no Fungi", and an e-value of 5. Finally, a total of 989 protein sequences were generated. We selected all syn-copalyl diphosphate synthase

sequences, (+)-copalyl diphosphate synthase sequences, and 100 ent-copalyl diphosphate synthase sequences. After removing the fragment sequences that are too short in length, selected sequences were finally used for sequence alignment. In addition, a search in the Uniprot database by the name syn-copalyl diphosphate synthase revealed that A0A8R7P944 and A0A8R7RGM7, which were not previously in sequence alignment, were also used for sequence alignment. The sequence alignment files were achieved in Align in Uniprot website. Consensus sequence logos were depicted using WebLogo 3[62].

**Reporting summary**. Further information on research design is available in the Nature Portfolio Reporting Summary linked to this article.

## Data availability

Cryo-EM density maps of OsCyc1 have been deposited in the Electron Microscopy Data Bank and the Protein Data Bank (PDB) under the following accession numbers: dimer (EMD-35207 and PDB ID: 8I6U), tetramer (EMD-35202 and PDB ID: 8I6P), hexamer (EMD-35206 and PDB ID: 8I6T) and OsCyc1$^{D367A}$ (EMD-35440 and PDB ID: 8IH5). The crystal structure was deposited in the Protein Data Bank (PDB) under the code 8KBW and is provided as Supplementary Data 1. The validation reports and the source data underlying the graphs and charts are included with this paper as Supplementary Data 2–7.

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

## Acknowledgements

We thank the staffs of the BL18U1 beamline at Shanghai Synchrotron Radiation Facility for technical assistance during data collection. We thank Ya Wang (Institute of Biophysics, Chinese Academy of Sciences) for assistance with crystal screen. We thank Xiaoxia Yu (Institute of Biophysics, Chinese Academy of Sciences) for assistance in analytical ultracentrifugation experiment and static light scattering analysis. We acknowledge Zhensheng Xie and Lili Niu for Mass spectrometry analysis at the Lab of Proteomics of Institute of Biophysics, Chinese Academy of Sciences. Cryo-EM data collection was carried out at the Center for Biological Imaging at the Institute of Biophysics (IBP), Chinese Academy of Sciences (CAS). We thank the Center for Biological Imaging (CBI), Institute of Biophysics, Chinese Academy of Science and would be grateful to Xujing Li, Boling Zhu, Lihong Chen, Deyin Fan and other staff members at the Center for Biological Imaging (IBP, CAS) for their support in data collection. We thank Wei Gao (School of Pharmaceutical Sciences, Capital Medical University) for providing the plasmids pMAL-c2X-TwCPS3 and pMAL-c2X-AtKS.

## Author contributions

X.M. and H.X. performed cloning, protein purification, crystallization, X-ray data collection and processing. X.M. performed the mutagenesis, GC-MS experiments, analytical ultracentrifugation experiments, cryo-EM sample preparation, cryo-EM data collection and processing, structure determination, refinement, docking and bioinformatics analysis. H.X. performed the static light scattering analysis. Y.T. and Y.L. assisted in the GC-MS experiments. Q.D. assisted in cryo-EM data processing. The manuscript was written by X.M. and T.J.

## Competing interests

The authors declare no competing interests.
