## [Peer Review File · Communications Chemistry]

Reviewers' comments:

Reviewer #1 (Remarks to the Author):

In this manuscript, the authors report the structure and function analysis of class 2 terpene cyclase OsCyc1 which generate syn-copalyl diphosphate (syn-CPP). The authors obtained X-ray crystallographic structure and cryo-EM structures of OsCyc1. The comparison of these structures as well as oligomeric analysis suggested that OsCyc1 mainly exists as tetramer, while the oligomerization is not essential for enzyme activity. The complex structure with GGPP, comparison of the active site architectures among CPP synthases, and following site-directed mutagenesis indicated the important residues for the enzyme activity. The 5 residues variant and 6 residues variant of OsCyc1 generated a new peak. This manuscript is well-written and easy to follow. The new structures of syn-CPP producing CPS would attract the interests of the researchers in the field of natural product chemistry and enzymology. Therefore, I would recommend it for acceptance with some revisions.

1. The authors need to determine the new product of OsCyc1 because this is a key result in this manuscript.
2. Figure 4; The panels in figure 4 are too small to see the details. Some enlarged views should be added in SI.
3. Line 202-203; Even though the sequence alignment did not provide significant insight, the results of sequence alignment should be shown in SI.
4. Line 224; The structure of syn-halima-5,13E-dienyl should be shown.
5. GC-MS charts for the enzyme reactions of all variants should be shown.
6. Supplementary Figure 6; please explain why the peak intensities of OsCyc1 are different in each panel.
7. The authors should discuss about the base residue(s) that abstract a hydrogen atom from labda-13E-en-8yl+ intermediate based on the docking model.

Reviewer #2 (Remarks to the Author):

In this manuscript, Kamada et al. present the first structures of copalyl diphosphate synthases (CPSs) that form syn-copalyl diphosphates (syn-CPP). Structures have been solved for CPSs that form other known stereochemically distinct products (+)-CPP and ent-CPP—yet no structures have been solved of syn-CPP and how slight enzymatic differences enable catalytic control. The authors therefore pursue understanding OsCyc1 which forms syn-CPP. Overall, while the authors seek intriguing questions of how syn-CPPs allow for stereospecific catalysis (and considering what features might lead to fine-tuning of stereospecificity), there are caveats in the structural biology experiments and conclusions drawn from these structures that make me hesitant to recommend this manuscript for publication without further experiments and clarification. Please see the following concerns:

Major concerns:

- I am primarily concerned about the conclusions drawn from mid-resolution structures without having access to coordinates/maps or figures in the paper displaying density, particularly showing amino acid residue density. The authors state early on that they will structurally focus on the ~3.5 angstrom crystal

structure. They then design experiments and evaluate structural implications based on this crystal structure. From my perspective, without seeing electron density and maps/coordinates confirming the conformation of individual residues, I would be hesitant to accept conclusions made at the atomic level concerning specific salt bridges, H-bonds (which probably should be written as “within H-bond distance”—although at this resolution it might be challenging to determine).

- Along these notes, I view the main conclusions of this paper to be focused on oligomeric state—studying oligomeric state would be completely understandable at this resolution. Anything further concluding how this structure provides mechanistic insight, atomic level amino acid residue interactions (especially for flexible residues), etc is questionable to me without viewing the electron density maps.

- I am confused about the relevance of the cryo-EM structures other than visualizing different oligomeric states. You classify dimers, tetramer, and hexamers on grids and solve 3D structures based on these classifications. Static light scattering and AUC experiments suggest that the oligomeric state in solution is likely between trimeric and tetrameric state. Why do you think you don't see any trimer on EM grids? Is the buffer/EM grid preparation conditions different? Could there be preferred orientation issues that are leading to a trimeric state not being observed—for example maybe the dimer model is lower resolution because it is averaging dimer and trimer particles.

- Admittedly areas of a crystal structure might be more resolved, particularly at more structured regions, such as at dimer interfaces. To evaluate this aspect, I would want to see figures exhibiting electron density, but specifically would like to see composite omit maps that show density for amino acid residues. This is critical to include at this resolution, where you are on the edge of being able to observe these features.

- Furthermore, as of now, the differences between R_{work} and R_{free} values of crystal structure are large (~0.07 difference) suggesting these structures have possibly been over-refined introducing bias. I would recommend working on refinement to have smaller gap between R_{work} and R_{free} to result in less biased maps.

- The protein B-factors for structures are very high! Again double check structure refinement. I think at this resolution, conclusions can be made particularly regarding oligomeric state, but I am concerned that further conclusions are being made beyond resolution limit (perhaps due to bias from refinement).

- Some experiments/conclusions drawn seemed unreasonable. For example, in paragraph 148-164, the authors discuss how disrupting a presumed S674/E677 interaction (is there density to support that interaction?) leads to primarily dimer formation instead of tetramer—they conclude they've made a mutation that leads to formation of a new dimer and the tetramer is not required for enzymatic activity; however, without structure information, the authors have not actually proven that an unobserved dimer as formed.

- Similarly in lines 165-170, positive surface residues are mutated under the presumption that perhaps positively charged residues in oligomeric state draw in and increase local concentration of negatively charged substrate—no difference in end product formation was observed. The authors thus include that oligomeric state has little impact on enzyme activity and drawing in substrate. These conclusions seem unfounded with data provided.

- Overall question I still had after this paper: what insight have the authors provided into why this OsCyc1 specifically produces syn-CPP vs other stereoisomers? Are such insights even possible at this resolution? Still open questions about what subtle enzymatic differences enable different product formation (several amino acids are suggested to play a role, but no conclusive evidence is found). While several combined mutations resulted in some minor additional product formation, questions

Minor concerns:

- Be careful referring to end-point assays (detecting product by GC-MS) as enzyme activity. Were assays performed on same time scale and under same conditions? Thus are product levels comparable?
- Lines 106-108. You are discussing how AUC of OsCyc166-767 exhibits “different oligomeric states”—this confused me because I was wondering different compared to what? I would recommend AUC experiments with wild-type OsCyc1 if you have not already done that. Where is the N-term of OsCyc1 located, and could it disrupt oligomeric state?
- Statements of “forming direct H-bond contacts” seem unreasonable at this resolution level. Can state that residues are likely within H bonding distance, based on probable residue conformation (if clear density is not observed).

Overall the data in this paper seem like they should be focused on how oligomeric state impacts activity—less conclusions at the atomic level can be made at this resolution. If such conclusions are going to be made, viewing density and conscious crystallographic refinement to avoid bias is essential—figures showing composite omit maps would strengthen any argument regarding mechanism or atomic level residue interactions. How do oligomeric states in syn-CPPs differ from (+)- and ent-CPPs? Do differences actually impact stereospecificity? Overall this is an interesting system, but I strongly recommend improved refinement to reduce bias and more direct figures showing density (and any limitations based on resolution). To review again, I would want access to coordinates/maps and figures showing electron density (particularly composite omit maps for areas including key residues).

Reviewer #3 (Remarks to the Author):

This manuscript describes structural analysis of the rice syn-CPP synthase, OsCyc1 (OsCPS4), with subsequent extensive mutagenesis attempting to uncover enzymatic structure-function relationships, particularly that underlying the somewhat unusual stereochemical outcome of the reaction catalyzed by this class II diterpene cyclase (DTC). Various structures were solved using x-ray crystallography or cryo-EM, none of which are particularly high-resolution, although sufficient to provide at least reasonably accurate side-chain configurations. Briefly, these can be summarized as xtal and cryo-EM structures of the wild-type enzyme and a cryo-EM structure of an inactive mutant (D367A) bound to the substrate GGPP. This was followed by mutagenesis first exploring the observed oligomerization as well as unusual product outcome. The work appears to be solid for the most part (although see below). However, the description and analysis needs improvement. For example, the utilized N-terminal truncation variant simply removes the plastid targeting sequence, representing a pseudo-mature DTC, as has been reported before (refs 6 & 7), and needs to be clarified here. More importantly, the structures could be mined for more information than currently presented. For example, the Asp acting as the catalytic acid is activated by not only the surrounding Asp but also a hydrogen bond to a conserved Asn, all of which needs to be described here. The GGPP bound structure also is not fully analyzed, with the suitability of the overall conformation of the substrate for the catalyzed reaction not mentioned, which suggests this is not true. For example, is it suitably positioned for cyclization? What is the distance between the

relevant carbons? Is the carbon to be protonated to initiate the reaction appropriately situated relative to the catalytic acid? What about with the docked syn-CPP product? Albeit this Asp is missing in the solved structure the methyl from the Ala replacement should still be near this carbon at least with substrate, if not also product. Unfortunately, the overlaid structures shown in Fig. 6d are not encouraging in this regard. The authors should be aware that none of the available DTC structures contain GGPP in the proper pre-catalytic conformation. Moreover, the lack of magnesium (Mg^{2+}) the substrate bound structure (both here and in the previously determined structures) is consistent with the observed conformation not being catalytically relevant. Note that no details are provided about the examination of Mg^{2+} effect on the assay, and previous rigorous studies strongly indicate that Mg^{2+} is essentially necessary (e.g., in the presence of chelating agents), in contrast to what is suggested here. While the extensive mutagenesis does lead to an extended (5 residue) mutant with some alternative product, this compound is not convincingly identified. More troublingly, the cited references for interpretation of the results do not pertain to DTCs, but the mechanistically distinct (class I) monoterpene synthases (ref. 31), or triterpene cyclases (ref. 19), which are mechanistically relevant (although this relationship is not presented here, leaving it confusing for the general reader). An interesting question with regards to this enzyme is the identity of the catalytic base, and it is puzzling that it is not really addressed here. It has been previously reported that the base might be H501, in a cited paper (ref. 17), but the positioning of this residue is not shown here. With regards to this question, the authors should note that even hydroxyl groups (e.g., from tyrosine, serine or threonine) can act as a catalytic base to deprotonate carbocations. Thus, the presence of such residues in the active site must be considered in this respect. In addition, particularly given the noted small cavity size of the active site, it seems worth noting that the H357W mutation may simply sterically block substrate binding, requiring other mutations to open it back up. Frankly, in the absence of more convincing evidence that the solved structures actually provide informative functional information, it is not clear that this study is of sufficient significance to justify publication in *Communications Chemistry* and is better suited to a more specialized journal such as *Biochemistry*. Regardless, the presentation needs significant improvement, as well as additional experimentation/analysis, as noted above.

Minor corrections:

Line 46: stemod-13(17)-ene is not bioactive

Line 53: delete "to"

Line 54: delete first "the"

Line 95: "the edge" should be "the outside edge" and please indicate active sites in Fig. 1c

Response to reviewers

We sincerely appreciate the reviewers for their thorough review and constructive comments on our manuscript entitled “Structural and functional investigations of *syn*-copalyl diphosphate synthase from *Oryza sativa*”. Below, we provide a point-to-point response to all of the comments and made necessary revisions in the manuscript accordingly. We hope that our revisions meet your expectations, and we are receptive to any further suggestions for improvement.

Reviewers' comments:

Reviewer #1 (Remarks to the Author):

In this manuscript, the authors report the structure and function analysis of class 2 terpene cyclase OsCyc1 which generate *syn*-copalyl diphosphate (*syn*-CPP). The authors obtained X-ray crystallographic structure and cryo-EM structures of OsCyc1. The comparison of these structures as well as oligomeric analysis suggested that OsCyc1 mainly exists as tetramer, while the oligomerization is not essential for enzyme activity. The complex structure with GGPP, comparison of the active site architectures among CPP synthases, and following site-directed mutagenesis indicated the important residues for the enzyme activity. The 5 residues variant and 6 residues variant of OsCyc1 generated a new peak. This manuscript is well-written and easy to follow. The new structures of *syn*-CPP producing CPS would attract the interests of the researchers in the field of natural product chemistry and enzymology. Therefore, I would recommend it for acceptance with some revisions.

Comment 1. The authors need to determine the new product of OsCyc1 because this is a key result in this manuscript.

Response: Thank you very much for your comments.

In order to identify the specific conformation of the product corresponding to peak 2 in the mass spectrum of OsCyc1^{5Mu}, we conducted further investigations and successfully identified the new product as *ent*-CPP.

We synthesized the codon-optimized AtKS gene for protein expression and elevated the amounts of both enzyme and substrate used in the product identification process. Detailed procedures can be found in the "Relative Enzyme Activity Assays" section of the Methods.

By comparing the retention time and mass spectrum of GC-MS, we confirmed that the specific conformation of the product corresponding to peak 2 in the mass spectrum of OsCyc1^{5Mu} is *ent*-CPP. Consequently, we have added the relevant results in the main

text of manuscript and added an additional figure in the Supplementary Figure 19.

The identification of the product undoubtedly enhances the content and novelty of our article, thereby improving the overall quality of our work. Thank you once again.

Comment 2. Figure 4; The panels in figure 4 are too small to see the details. Some enlarged views should be added in SI.

Response: Thank you very much for your constructive comment. We sincerely appreciate your careful attention to Figure 4 and your suggestion to provide enlarged views for better clarity.

We have enlarged specific sections of Figure 4 that may appear unclear and included the magnified view of these sections in the Supplementary Figure 12.

Comment 3. Line 202-203; Even though the sequence alignment did not provide significant insight, the results of sequence alignment should be shown in SI.

Response: Thank you very much for your valuable and constructive feedback. We have taken your comments into consideration and have added two Supplementary Figures containing additional sequence alignment results as Supplementary Figure 15 and 16.

In Supplementary Figure 15, our focused was on class II diterpene synthases that utilize GGPP as a substrate to produce *syn*-CPP, *ent*-CPP, (+)-CPP, (+)-8-hydroxycopalyl diphosphate, *ent*-8-hydroxycopalyl diphosphate and peregrinol diphosphate. This alignment enables a comprehensive comparison of various enzymes involved in diterpene synthase.

Furthermore, in Supplementary Figure 16, we have included a sequence conservation analysis of 8-hydroxycopalyl diphosphate synthases. These sequences for alignment were obtained from the Uniprot website using relevant keywords such as copal-8-ol diphosphate hydratase, 8-hydroxycopalyl diphosphate synthase, and 8-LPP. Subsequently, sequences were selected based on their record names, with sequence lengths greater than 470 amino acids and domains containing β and γ domains. Moreover, we also added CfTPS2 (UniProtKB: X4ZWN5, diterpene synthase TPS2) and AbCAS (UniProtKB: H8ZM73, bifunctional cis-abienol synthase) to the alignment sequences. The sequence conservation analysis was performed using the WebLogo3 website.

Additionally, we have highlighted the specific amino acids comprising the active site of OsCyc1 in Supplementary Figure 15 and Supplementary Figure 16. These amino acids include those near the pyrophosphate group (M194, E199, K233, N405, K453) and those near the isoprene tail (V196, T248, H251, H275, C310, I311, L314, Y317, H357, V363, D365, D367, D368, C399, L400, W454, H501). Besides, we have provided the molecular structure of (+)-8-hydroxycopalyl diphosphate, *ent*-8-hydroxycopalyl diphosphate and peregrinol diphosphate in Supplementary Figure 15c.

We hope that these additional materials will provide clarity to readers and address the concerns raised by the reviewers. Thank you once again for your valuable comment, which has greatly contributed to the enhancement of our work.

Comment 4. Line 224; The structure of *syn*-halima-5,13E-dienyl should be shown.

Response: Thank you very much for your advice. We have incorporated the structure of *syn*-halimadienyl diphosphate (*syn*-halima-5,13E-dienyl diphosphate) into Supplementary Figure 15c. Additionally, we have cited this picture after the following description on Page 6, line 254-256:

“In previous report, substitution of H501 in OsCyc1 has been well studied showing that H501A, H501D and H501F produce *syn*-halimadienyl diphosphate in an *E.coli* modular metabolic engineering system (Supplementary Figure 15c).”

Comment 5. GC-MS charts for the enzyme reactions of all variants should be shown.

Response: Thank you very much for your feedback. We appreciate your valuable comments. We agree that displaying GC-MS charts of all variants used for the enzyme activity reactions would provide clarity for readers. As a result, we have included the relevant constructs in the supplementary materials, specifically in Extended Data Table 3, Supplementary Figures 2, 4, 10, 11, 17-19.

The GC-MS experiments were conducted in various runs, and due to variations in the length of the DB-5MS column of each batch, there were slight differences in the peak positions of *syn*-CPP. In each experiment, we used *syn*-CPP generated by OsCyc1 as a reference standard for comparison.

Comment 6. Supplementary Figure 6; please explain why the peak intensities of OsCyc1 are different in each panel.

Response: Thank you very much for your comments.

In Supplementary Figure 6 (Supplementary Figure 18 of the revised version), the experiments were not conducted from the same batch. The standard samples used in each experiment, some of which were frozen and stored in the refrigerator, may have experienced volatility or degradation. As a result, there are variations in the intensities of the standard samples in the GC-MS experimental results.

Furthermore, disparities in the peak positions of the standard samples occur for the following reasons:

The mass spectrometry equipment was shared among multiple users, and similarly, the DB-5MS column used in this study was also shared. The mutants involved in this study were not all analyzed in the same batch. For each mass spectrometry detection, a new

column installation was required, which resulted in a partial truncation of the column. As a result, the column would become progressively shorter with each use, leading to slight deviations in the peak positions during each detection. To account for this, a standard reference was used for comparison in each mass spectrometry experiment. For information regarding the truncation of the column during installation, reference can be found on the Agilent website at the following link:

<https://www.agilent.com.cn/en/products/gas-chromatography/installgccolumn>.

To avoid any confusion for readers, we have provided explanations in the figure legends of the corresponding GC-MS experimental results as the following:

"The GC-MS experiments were conducted in various runs, and due to differences in the length of the DB-5MS column in each batch, there were slight differences in the peak positions of *syn*-CPP."

Specific figure legends can be found in Supplementary Figures 10, 11, 17 and 18.

Comment 7. The authors should discuss about the base residue(s) that abstract a hydrogen atom from λ -13E-en-8yl⁺ intermediate based on the docking model.

Response: Thank you very much for your suggestion.

In order to analyze which base residue that accepts the hydrogen atom in the end, we performed molecular docking of *syn*-CPP with OsCyc1^{D367A}. The results showed that H501 and W495 were in close proximity to the C17 and C8 atoms where the λ -13E-en-8-yl⁺ carbon cation was ultimately quenched. Taking into account previous report on H501 mutation experiments, we speculate that H501 may be the base residue that ultimately accepts the hydrogen atom. Therefore, we have added Supplementary Figure 20 and updated Figure 6, and included the following discussion in Page 8, Line 336-343:

"Interestingly, in the molecular docking results of *syn*-CPP and OsCyc1^{D367A}, Y317, W495, and H501 are close to the C17 atom of *syn*-CPP (Figure 6d). These amino acids may be involved in the carbon cation quenching of the intermediate λ -13E-en-8-yl⁺ (Figures 6d, 6f). Previous report showed that mutations of H501 to aspartic acid or phenylalanine can generate halimadienyl diphosphate¹⁷. Halimadienyl diphosphate is quenched from the intermediate λ -13E-en-10-yl⁺, which is derived from the carbon cation transfer of the intermediate λ -13E-en-8-yl⁺³⁷ (Supplementary Figure 20c). Based on these findings, we speculate that H501 may be the final base residue that extracts the proton from λ -13E-en-8-yl⁺."

Reviewer #2 (Remarks to the Author):

In this manuscript, Kamada et al. present the first structures of copalyl diphosphate synthases (CPSs) that form *syn*-copalyl diphosphates (*syn*-CPP). Structures have been

solved for CPPs that form other known stereochemically distinct products (+)-CPP and ent-CPP—yet no structures have been solved of syn-CPP and how slight enzymatic differences enable catalytic control. The authors therefore pursue understanding OsCyc1 which forms syn-CPP. Overall, while the authors seek intriguing questions of how syn-CPPs allow for stereospecific catalysis (and considering what features might lead to fine-tuning of stereospecificity), there are caveats in the structural biology experiments and conclusions drawn from these structures that make me hesitant to recommend this manuscript for publication without further experiments and clarification. Please see the following concerns:

Major concerns:

Comment 1.- I am primarily concerned about the conclusions drawn from mid-resolution structures without having access to coordinates/maps or figures in the paper displaying density, particularly showing amino acid residue density. The authors state early on that they will structurally focus on the ~3.5 angstrom crystal structure. They then design experiments and evaluate structural implications based on this crystal structure. From my perspective, without seeing electron density and maps/coordinates confirming the conformation of individual residues, I would be hesitant to accept conclusions made at the atomic level concerning specific salt bridges, H-bonds (which probably should be written as “within H-bond distance”—although at this resolution it might be challenging to determine).

Response: Thank you very much for your valuable comments. The insights and suggestions you have provided are of immense significance to us, and we genuinely appreciate the constructive nature of your feedback.

In order to address any potential uncertainties and to offer readers a clearer understanding of our structural information, we have added Supplementary Figures 5 to 9, which depict the amino acid densities involved in protomer interactions.

Our crystal structure was solved at a resolution of 3.5 Å, and Cryo-EM hexamer structures of both WT and OsCyc1^{D367A} were solved at resolutions of 3.7 Å and 4 Å, respectively. Notably, the cryo-EM structure of OsCyc1^{D367A} showcases more distinct densities for side chains at the protomer interaction sites. As a result, these newly added images display the 2mFo-DFc map and composite omit map for the crystal structure, along with the Cryo-EM map of the OsCyc1^{D367A} structure.

We have labeled the bond lengths between interacting amino acids that show clear side chain densities. For interacting amino acids with well-defined bond lengths, we have annotated their specific interactions, such as the explicit hydrogen bond between S674 and E677 in Supplementary Figure 7b. In the case of amino acids with unclear side chain densities, we have speculated about their potential interactions in the manuscript. Please refer to Section "Biochemical Analysis of the OsCyc1 Oligomer State" for the revisions made in the manuscript (Page 4, Line 136-164).

Comment 2- Along these notes, I view the main conclusions of this paper to be focused on oligomeric state—studying oligomeric state would be completely understandable at this resolution. Anything further concluding how this structure provides mechanistic insight, atomic level amino acid residue interactions (especially for flexible residues), etc is questionable to me without viewing the electron density maps.

Response: Thank you very much for your comments.

In addition to the inclusion of Supplementary Figures 5-9 as mentioned in **Comment 1**, we have also added additional density maps of the interacting amino acids mentioned in the manuscript. Specifically, the detailed density maps corresponding to active site have been included as Supplementary Figure 14. We hope that these density maps will help alleviate your concerns. Once again, we sincerely appreciate your valuable suggestions.

Comment 3- I am confused about the relevance of the cryo-EM structures other than visualizing different oligomeric states. You classify dimers, tetramer, and hexamers on grids and solve 3D structures based on these classifications. Static light scattering and AUC experiments suggest that the oligomeric state in solution is likely between trimeric and tetrameric state. Why do you think you don't see any trimer on EM grids? Is the buffer/EM grid preparation conditions different? Could there be preferred orientation issues that are leading to a trimeric state not being observe—for example maybe the dimer model is lower resolution because it is averaging dimer and trimer particles.

Response: Thank you very much for your comments.

The static light scattering experiment indicates that the molecular weight of OsCyc1 lies between that of a trimer and a tetramer. Analytical ultracentrifugation analysis not only suggests that the molecular weight of OsCyc1 falls within the range of a trimer and a tetramer, but also indicates that OsCyc1 primarily adopts a specific state in solution, specifically either trimer or tetramer. Considering that tetrameric particles dominate in our cryo-electron microscopy structure determination, we hypothesize that OsCyc1 predominantly exists in a tetrameric state in solution.

During the process of electron microscopy structure determination, the selected particles underwent 2D and 3D classifications. However, distinguishable trimeric particles were not observed under this condition. The buffer used in the electron microscopy sample preparation was consistent with that used for static light scattering and analytical ultracentrifugation experiments.

Your suggestion regarding the dimer model's lower resolution because it averages dimer and trimer particles is valuable. We cannot entirely rule out the possibility of trimeric state in solution. Therefore, in Page 3, Line 114-115 of the manuscript, the phrase "strongly suggest" was changed to "imply". Taken these together, we revised the sentences as "Combined with the cryo-EM structure results, these findings imply that

the major oligomeric form of OsCyc1 in solution is tetrameric.”.

Comment 4- Admittedly areas of a crystal structure might be more resolved, particularly at more structured regions, such as at dimer interfaces. To evaluate this aspect, I would want to see figures exhibiting electron density, but specifically would like to see composite omit maps that show density for amino acid residues. This is critical to include at this resolution, where you are on the edge of being able to observe these features.

Response: Thank you for your valuable feedback. We greatly appreciate the time and effort you've dedicated to reviewing our work.

In response to your concerns, we have added Supplementary Figures 5 to 9, which display composite omit maps specifically highlighting the interacting amino acids of protomers in the crystal structure. We hope that these additional images will significantly enhance the overall quality and comprehensibility of our manuscript. We sincerely hope that these updates will alleviate any uncertainties you had. Once again, we thank you for your meticulous review and constructive feedback.

Comment 5- Furthermore, as of now, the differences between R_{work} and R_{free} values of crystal structure are large (~0.07 difference) suggesting these structures have possibly been over-refined introducing bias. I would recommend working on refinement to have smaller gap between R_{work} and R_{free} to result in less biased maps.

Response: Thank you very much for your comments.

We have re-refined the crystal structure and submitted it to the PDB website, with the accession code 8KBW. The refinement parameters of the revised crystal structure can be found in Extended Data Table 1, with $R_{work} = 0.2014$ and $R_{free} = 0.2630$. We hope these changes can address any concerns you had. Once again, we sincerely appreciate your valuable and constructive feedback.

Comment 6- The protein B-factors for structures are very high! Again double check structure refinement. I think at this resolution, conclusions can be made particularly regarding oligomeric state, but I am concerned that further conclusions are being made beyond resolution limit (perhaps due to bias from refinement).

Response: Thank you very much for your comments.

As mentioned above, we have re-refined the crystal structure and submitted it to the PDB website, with the accession code 8KBW. The refinement parameters of the revised crystal structure can be found in Extended Data Table 1, with an average B-factor dropped from 95.45 to 81.33.

Comment 7- Some experiments/conclusions drawn seemed unreasonable. For example, in paragraph 148-164, the authors discuss how disrupting a presumed S674/E677 interaction (is there density to support that interaction?) leads to primarily dimer formation instead of tetramer—they conclude they've made a mutation that leads to formation of a new dimer and the tetramer is not required for enzymatic activity; however, without structure information, the authors have not actually proven that an unobserved dimer as formed.

Response: Thank you very much for your comments. In Supplementary Figures 5 to 9, we have provided additional 2mFo-DFc map and composite omit map for the crystal structure, as well as the Cryo-EM map of the OsCyc1^{D367A} structure. In Supplementary Figure 7b, densities corresponding to S674 and E677 are clearly visible. In both the crystal structure and the cryo-electron microscopy structure, the distance between S674 and E677 is approximately 2.5 Å. Based on this observation, we propose that a hydrogen bond forms between these two amino acids.

The molecular weight of S674A/E677A lies between that of a monomer and a dimer. Your suggestion holds considerable value – without direct structural evidence, we cannot definitively classify it as a dimer. Therefore, we have revised the manuscript accordingly in Page 5, Line 179-181 as the following:

“Considering the results of our size exclusion chromatography experiments, we hypothesized that S674A/E677A disrupts the oligomeric state of wild-type OsCyc1.”

We genuinely appreciate your insightful comments and constructive feedback again.

Comment 8- Similarly in lines 165-170, positive surface residues are mutated under the presumption that perhaps positively charged residues in oligomeric state draw in and increase local concentration of negatively charged substrate—no difference in end product formation was observed. The authors thus include that oligomeric state has little impact on enzyme activity and drawing in substrate. These conclusions seem unfounded with data provided.

Response: Thank you very much for your comments. Your suggestions are indeed constructive. Our initial intention was to convey that S674A/E677A mutant maintains relative enzyme activity similar to the wild type, implying that tetramer formation seems to have minimal impact on enzyme activity. The results of the mutations at surface positive residues (not affecting enzyme activity) serve as supplementary experiments to the outcomes of the S674A/E677A mutation.

To convey this viewpoint more clearly, we have made the following revisions in Page 5, Line 186 as the following:

“However, the replacement of these residues with alanine showed a little effect on enzyme activity (Fig. 3a and Supplementary Fig. 11), which is in agreement with the result from the S674A/E677A mutant, showing that the tetramer seems have a minor impact on enzyme activity *in vitro*.”

We hope these explanations address your concerns. Once again, we sincerely appreciate your valuable and constructive feedback.

Comment 9- Overall question I still had after this paper: what insight have the authors provided into why this OsCyc1 specifically produces *syn*-CPP vs other stereoisomers? Are such insights even possible at this resolution? Still open questions about what subtle enzymatic differences enable different product formation (several amino acids are suggested to play a role, but no conclusive evidence is found). While several combined mutations resulted in some minor additional product formation, questions

Response: Thank you very much for your insightful and constructive comments.

We have analyzed the structures of both *apo* and ligand-bound OsCyc1, and from our structural analysis, we deduced that the active site of OsCyc1 is the most compact among the three diterpene synthases that generate stereochemically different CPPs. The formation of CPPs share the same intermediate λ -13*E*-en-8-yl⁺. The difference lies in the conformation of the decalin ring of the intermediates. While the decalin ring in both (+)-CPP and *ent*-CPP adopt chair-chair conformations, the decalin ring of *syn*-CPP folds into a chair-boat conformation, which is the most unstable. The compactness of the active site of OsCyc1 might facilitate the folding of the substrate GGPP into the less stable chair-boat conformation required for *syn*-CPP formation. This perspective has been discussed in our manuscript in the section of Discussion (Page 9, Line 362-366).

We have included density maps of the active site residues of OsCyc1^{D367A} in the Supplementary Figure 14. Our Cryo-EM structures exhibit clear visibility of amino acids side chains.

Through both structural and mutational experiments, we have confirmed that the residues H275, I311, L314, Y317, and H357 collectively contribute to the formation of the *syn*-CPP conformation. Mutating these residues can alter the shape of the active site pocket. From molecular docking results we can infer that I311 and H275 contribute to an active site pocket shape unfavorable for other CPP conformations, thereby ensuring the production of *syn*-CPP.

Furthermore, our molecular docking results are consistent with a previous report by Potter et al. (Product Rearrangement from Altering a Single Residue in the Rice *syn*-Copalyl Diphosphate Synthase. *Organic Letters* 18, 1060-1063, 2016), and now we added the discussion that H501 may serve as the final residue to accept a proton. These details have been added to the "Product Stereochemistry Change Through Site-Directed Mutagenesis" section (Page 8, Line336-343).

Minor concerns:

Comment 10- Be careful referring to end-point assays (detecting product by GC-MS)

as enzyme activity. Were assays performed on same time scale and under same conditions? Thus are product levels comparable?

Response: Thank you very much for your comments.

The term "enzyme activity" mentioned in our paper should be changed to "relative enzyme activity," and we have made the necessary correction in the manuscript.

In the “Relative Enzyme Activity Assays” section under “Materials and Method”, we have provided detailed descriptions of the enzyme activity assay procedure.

Comment 11- Lines 106-108. You are discussing how AUC of OsCyc166-767 exhibits “different oligomeric states”—this confused me because I was wondering different compared to what? I would recommend AUC experiments with wild-type OsCyc1 if you have not already done that. Where is the N-term of OsCyc1 located, and could it disrupt oligomeric state?

Response: Thank you very much for your comments.

I apologize for any confusion caused. The term "different" should indeed be replaced with "various," and we have made the necessary correction in the manuscript.

We greatly appreciate your feedback and suggestions. We will take your recommendations into careful consideration, and in our upcoming research, we plan to conduct analytical ultracentrifugation experiments for the wild-type OsCyc1.

Comment 12- Statements of “forming direct H-bond contacts” seem unreasonable at this resolution level. Can state that residues are likely within H bonding difference, based on probable residue conformation (if clear density is not observed).

Response: Thank you very much for your suggestions. Following your advice, we have made the necessary revisions in the main text. For example:

“Residues located within or near $\alpha 22$ interact with residues in $\alpha 25$, such as Q627 with R720, which are at a hydrogen bonding distance in the crystal structure (Fig. 2b and Supplementary Fig 5a, 5b).” (Page 4, Line136-138)

Comment 13 Overall the data in this paper seem like they should be focused on how oligomeric state impacts activity—less conclusions at the atomic level can be made at this resolution. If such conclusions are going to be made, viewing density and conscious crystallographic refinement to avoid bias is essential—figures showing composite omit maps would strengthen any argument regarding mechanism or atomic level residue interactions. How do oligomeric states in syn-CPPs differ from (+)- and ent-CPPs? Do differences actually impact stereospecificity? Overall this is an interesting system, but I strongly recommend improved refinement to reduce bias and more direct figures

showing density (and any limitations based on resolution). To review again, I would want access to coordinates/maps and figures showing electron density (particularly composite omit maps for areas including key residues).

Response: Thank you very much for your suggestions. We have added density maps for the relevant amino acids in the Supplementary Figure 5-9. Currently, there is no literature available regarding the oligomeric states of (+)-CPSs and *ent*-CPSs in plants.

Reviewer #3 (Remarks to the Author):

This manuscript describes structural analysis of the rice syn-CPP synthase, OsCyc1 (OsCPS4), with subsequent extensive mutagenesis attempting to uncover enzymatic structure-function relationships, particularly that underlying the somewhat unusual stereochemical outcome of the reaction catalyzed by this class II diterpene cyclase (DTC). Various structures were solved using x-ray crystallography or cryo-EM, none of which are particularly high-resolution, although sufficient to provide at least reasonably accurate side-chain configurations. Briefly, these can be summarized as xtal and cryo-EM structures of the wild-type enzyme and a cryo-EM structure of an inactive mutant (D367A) bound to the substrate GGPP. This was followed by mutagenesis first exploring the observed oligomerization as well as unusual product outcome. The work appears to be solid for the most part (although see below).

Comment 1. However, the description and analysis needs improvement. For example, the utilized N-terminal truncation variant simply removes the plastid targeting sequence, representing a pseudo-mature DTC, as has been reported before (refs 6 & 7), and needs to be clarified here.

Response: Thank you very much for your comments. The purpose of conducting the N-terminal truncation was to obtain a stable protein for the purpose of structural analysis. The relevance sentence has been changed to:

“To determine the structure of OsCyc1, we used the stable N-terminal truncation variant of OsCyc1, OsCyc1⁶⁹⁻⁷⁶⁷, which was catalytically active as shown by our relative enzyme activity assay (Supplementary Fig. 1 and 2).” (Page 2, Line 82-84)

Comment 2. More importantly, the structures could be mined for more information than currently presented. For example, the Asp acting as the catalytic acid is activated by not only the surrounding Asp but also a hydrogen bond to a conserved Asn, all of which needs to be described here.

Response: Thank you very much for your comments.

We have added the structural information concerning the DXDD motif and the conserved asparagine N414 in the "Structure of GGPP in Syn-Copalyl Diphosphate

Synthase OsCyc1" section as the following (Page 5-6, Line 212-222):

“The DXDD motif and the conserved asparagine N414 are situated at the terminus of the isoprene tail of GGPP in the OsCyc1^{D367A} structure^{23, 26} (Supplementary Fig. 15). The distance between D365 and the isoprene tail of GGPP is 3.5 Å, whereas the measurement of the distance between the unmutated D367 and the substrate is not feasible due to its mutation to alanine (Supplementary Fig. 14c, 14d). In the crystal structure of OsCyc1, the distance between D367 and the conserved N414 is 2.97 Å (Supplementary Fig. 14e, 14f). Upon superimposing the crystal structure and the OsCyc1^{D367A} structure, the measured distance between D367A and GGPP is 2.82 Å, which is suitable for initiating the cyclization reaction (Supplementary Fig. 14g). However, in the OsCyc1^{D367A} structure, GGPP adopts a linear conformation, possibly reflecting an early stage of the reaction, which might not be represent the cyclization state.”

Besides, Supplementary Figure 14 has been included to provide additional details about the densities.

We sincerely value your insightful and constructive feedback.

Comment 3. The GGPP bound structure also is not fully analyzed, with the suitability of the overall conformation of the substrate for the catalyzed reaction not mentioned, which suggests this is not true. For example, is it suitably positioned for cyclization? What is the distance between the relevant carbons? Is the carbon to be protonated to initiate the reaction appropriate situated relative to the catalytic acid? What about with the docked *syn*-CPP product? Albeit this Asp is missing in the solved structure the methyl from the Ala replacement should still be near this carbon at least with substrate, if not also product. Unfortunately, the overlaid structures shown in Fig. 6d are not encouraging in this regard. The authors should be aware that none of the available DTC structures contain GGPP in the proper pre-catalytic conformation.

Response: Thank you very much for your comments.

The linear conformation of substrate GGPP in the structure is not conducive to the cyclization reaction. We have introduced relevant discussion in the "Structure of GGPP in *Syn*-Copalyl Diphosphate Synthase OsCyc1" section. Additionally, we have superimposed the crystal structure with the OsCyc1^{D367A} structure, measured the distance between the isoprene tail of GGPP and the corresponding amino acids in the protein, and included this information in Supplementary Figure 14. Based on the distance information, this proximity is suitable for initiating the reaction. However, since GGPP is linear and not folded into the corresponding cyclic structure, it does not represent the reaction state. Please refer to Page 5-6, Line 212-222 for further details.

Furthermore, we have conducted molecular docking of *syn*-CPP with OsCyc1^{D367A}. The results can be found in Figure 6 and Page 8, Line 336-343 as following:

“Interestingly, in the molecular docking results of *syn*-CPP and OsCyc1^{D367A}, Y317, W495, and H501 are close to the C17 atom of *syn*-CPP (Figure 6d). These amino acids

may be involved in the carbon cation quenching of the intermediate lambda-13*E*-en-8-yl⁺ (Figures 6d, 6f). Previous report showed that mutations of H501 to aspartic acid or phenylalanine can generate halimadienyl diphosphate¹⁷. Halimadienyl diphosphate is quenched from the intermediate halima-13*E*-en-10-yl⁺, which is derived from the carbon cation transfer of the intermediate lambda-13*E*-en-8-yl⁺³⁷ (Supplementary Figure 20c). Based on these findings, we speculate that H501 may be the final base residue that extracts the proton from lambda-13*E*-en-8-yl⁺.”

Comment 4. Moreover, the lack of magnesium (Mg²⁺) the substrate bound structure (both here and in the previously determined structures) is consistent with the observed conformation not being catalytically relevant. Note that no details are provided about the examination of Mg²⁺ effect on the assay, and previous rigorous studies strongly indicate that Mg²⁺ is essentially necessary (e.g., in the presence of chelating agents), in contrast to what is suggested here.

Response: Thank you very much for your review and suggestions!

Based on your advice, we conducted new experiments. In the previous experiments, various magnesium ions were added to the reaction buffer A. In the new experiments, we added 1 mM, 5 mM, and 10 mM EDTA to chelate magnesium ions in buffer A, and measured the relative enzyme activity in the presence of different EDTA concentrations. The results showed a significant reduction in enzyme activity upon EDTA addition. For more detailed information, please refer to Supplementary Figure 4 and Page 3, Line 122-126 as following:

“In the absence of additional magnesium ions in the solution, OsCyc1 demonstrates its highest level of activity. The presence of extra magnesium ions partially inhibits the enzyme's activity (Supplementary Fig. 2 and 4b). However, when EDTA is used to chelate potential magnesium ions, the activity of OsCyc1 is inhibited by almost 90% (Supplementary Fig. 4c, 4e).”

Comment 5. While the extensive mutagenesis does lead to an extended (5 residue) mutant with some alternative product, this compound is not convincingly identified. More troublingly, the cited references for interpretation of the results do not pertain to DTCs, but the mechanistically distinct (class I) monoterpene synthases (ref. 31), or triterpene cyclases (ref. 19), which are mechanistically relevant (although this relationship is not presented here, leaving it confusing for the general reader).

Response: Thank you very much for your comments.

In order to address the concerns of the reviewer, we have conducted extensive work to identify the products of OsCyc1^{5Mu}. We synthesized a codon-optimized gene of AtKS and purified high-quality protein of AtKS for subsequent enzyme reactions. Additionally, we increased the amount of GGPP in the reaction system to 100 μg to enhance substrate conversion. Through these efforts, we successfully identified the new

product as *ent*-CPP. We have updated the results in the main text, made modifications to Figure 6, and supplemented new picture in Supplementary Figure 19.

The reason we cited literature related to both monoterpenes and triterpenes is because terpene synthase reactions share many common features. We acknowledge the need to clarify this connection for readers. Therefore, in the introduction section of our manuscript at Page2, Line 47-53, we have added the following explanation:

“The catalytic mechanisms of terpene synthases share several common characteristics. Catalytic reactions of terpene synthases are multistep cascade reactions with multiple carbocation intermediates and carbocation rearrangements occur through a 1,2-hydride shift and/or alkyl shift to form new skeleton backbone products¹⁵. Class II terpene synthases have the characteristic DXDD motif in which the middle aspartic acid generally acts as the catalytic acid that initiates the reaction by protonation, and the catalytic base is variable^{16, 17}. The mechanisms underlying class II diTPSs resemble other terpene synthase reaction mechanisms, but also have their own characteristics.”

We hope that these explanations address your concerns and those of the readers. Once again, we sincerely appreciate your review of our article.

Comment 6. An interesting question with regards to this enzyme is the identity of the catalytic base, and it puzzling that it is not really addressed here. It has been previously reported that the base might be H501, in a cited paper (ref. 17), but the positioning of this residue is not shown here. With regards to this question, the authors should note that even hydroxyl groups (e.g., from tyrosine, serine or threonine) can act as a catalytic base to deprotonate carbocations. Thus, the presence of such residues in the active site must be considered in this respect.

Response: Thank you very much for your feedback.

We appreciate the insightful question you have raised, as it indeed addresses a crucial aspect of the catalytic mechanism. In order to identify the catalytic base, we conducted molecular docking using *syn*-CPP and OsCyc1^{D367A}. The docking results revealed that H501, W495 and Y317 are situated near C17 of *syn*-CPP, where the carbon carbocation of intermediate lambda-13*E*-en-8-yl⁺ would be quenched. Considering prior literature reporting that the H501 mutant variant can generate halimadienyl diphosphate, whose precursor is halima-13*E*-en-10-yl⁺, the subsequent product after the transfer of the carbon carbocation of intermediate lambda-13*E*-en-8-yl⁺. Based on these findings, we hypothesized that the H501 mutation disrupts the quenching of intermediate lambda-13*E*-en-8-yl⁺, leading to the formation of halimadienyl diphosphate. Consequently, we propose that H501 may function as the catalytic base. We have included a discussion on this issue in our manuscript (Page 8, Line 336-343), along with the revised Figure 6, and Supplementary Figure 20.

Comment 7. In addition, particularly given the noted small cavity size of the active site,

it seems worth noting that the H357W mutation may simply sterically block substrate binding, requiring other mutations to open it back up.

Response: Thank you very much for your comments.

Yes, you're right. The shape and size of the pocket play a crucial role in determining the different stereochemical conformations of CPPs.

Comment 8. Frankly, in the absence of more convincing evidence that the solved structures actually provide informative functional information, it is not clear that this study is of sufficient significance to justify publication in Communications Chemistry and is better suited to a more specialized journal such as Biochemistry. Regardless, the presentation needs significant improvement, as well as additional experimentation/analysis, as noted above.

Response: Thank you very much for your comments. We have conducted additional experiments and analysis to improve the quality of the manuscript. We hope that these efforts will enhance the overall quality of the paper. Once again, we appreciate your thorough review.

Minor corrections:

Line 46: stemod-13(17)-ene is not bioactive

Line 53: delete “to”

Line 54: delete first “the”

Line 95: “the edge” should be “the outside edge” and please indicate active sites in Fig. 1c

Response: Thank you for your valuable feedback. Your suggestions are greatly appreciated, and we have already made the necessary changes.

Reviewers' comments:

Reviewer #1 (Remarks to the Author):

The manuscript has been modified according to the reviewers' comments that were properly addressed.

Reviewer #2 (Remarks to the Author):

In this manuscript, the authors present the first structures of CPPs that form syn-CPP and through biochemical and structural analysis, identifying residues that play a role in dictating stereochemistry of this product, focusing on OsCyc1. Furthermore, they explore how oligomeric state impacts activity, with OsCyc1 exhibiting a tetrameric state in solution vs other terpenoid synthases that are typically dimers or monomers in solution. Overall I think the authors have greatly strengthened this manuscript with supporting figures that show electron density (including composite omit maps from crystal structure) and clearly conveying both where strong electron density is observed and regions where side chain density is weaker and limited—these edits better allows the reader to know what conclusions can be drawn from these data. Furthermore, I appreciate the authors thorough responses to my major concerns. I am happy to recommend this manuscript for publication with minor revisions, clarification, and suggestions (see below):

Questions for clarification:

1. The authors added a cryo-EM structure of OsCyc1D367A in its hexameric form. For the wild-type structure, tetramers were the predominant species. Were hexamers the predominant oligomeric form for this variant, and if so, do the authors have any speculation as to what might have caused this difference? Were grids prepared the same way? Authors might want to comment on this aspect since they emphasize how OsCyc1 predominantly forms tetramers in solution.
2. I was surprised by the experimental result that adding EDTA to OsCyc1 relative enzyme activity assays significantly inhibiting activity (Supplemental Fig 2 and 4b). If Mg²⁺ is proposed to partially inhibit relative activity, and highest activity is observed in the absence of additional Mg²⁺ ions, I would have hypothesized that adding EDTA to chelate additional Mg²⁺ would have restored some activity. I would recommend authors comment on this result of why EDTA strongly inhibits relative activity if Mg²⁺ is not essential.
3. Lines 325-328: “These results indicated that H275 plays an important role in syn-CPP production. This speculation is in line with the comprehensive bioinformatics analysis of syn-CPSs, ent-CPSs and (+)-CPSs in plants, which shows that H275 is always histidine and different from other CPSs (Fig. 6e and Supplementary Fig. 21-23).” I was confused by the wording “H275 is always histidine”. It seems like you mean it is strictly conserved, but in the figures, it seems that despite being largely conserved, in some sequences, it is another residue (difficult to see in sequence logos what other residues are found at that position and whether they have similar chemical properties).
4. Lines 340-347: You mention how these residues including H501 are close to the C17 atom, and propose H501 to potentially act as a catalytic base. In the structure, the distance of H501 to C17 is 5 angstroms. Based on results and rationale, it does appear H501 probably plays an important role, but this distance seems far for a proton transfer—I wonder whether there is a closer potential base (another

residue or perhaps even water acting as a base, activated by a nearby residue).

5. The authors did put substantial effort into improving refinement of the structure. To address the gap between R_{work} and R_{free} , I am wondering if the authors have tried optimizing target X-ray/ADP or X-ray/stereochemistry weights during computational refinement? This can often help shrink that gap/reduce bias if model has become overrefined. I think it is better to have a slightly higher R_{work} , but smaller gap between R_{work} and R_{free} , if possible to minimize bias.

Minor typos and suggestions:

1. When aligning structures of OsCyc1 with other CPPs and making statements of high similarity, including a quantitative RMSD would be helpful (for example, authors state in lines 98-100 that OsCyc1 is structurally similar to AtCPS, AgAS, and SmMDS—including RMSDs here or in the figure legend would be helpful).
2. Lines 142-144: “The interactions between α_{23} and α_{24} include R652 with Q675, as well as E640 with K686, possibly through van der Waals interactions (Fig. 2c and Supplementary Fig. 6a-6c).” Would you expect those to only be van der Waals interactions or possibly a salt bridge between Glu and Lys?
3. Lines 149-150, change “particles number” to “particle numbers”. In addition, I would refer the reader to supplemental table that shows particle number or even include in main text to fully convey this point.
4. Lines 190-191: change to “showing that the tetramer seems to have a minor impact...”
5. Lines 213-214: “In the structure of OsCyc1, K233, N405 and K453 interacts with the pyrophosphate, while M194 and E199 are close to the pyrophosphate of GGPP”. I was confused by this sentence—I would use more descriptive language for “close to” vs “interacts”. Are you saying that M194 and E199 might interact with the pyrophosphate—especially with E199 being a negatively charged group, would you expect it to interact with a negatively charged substrate moiety?
6. Lines 242-244: “Remarkably, compared to the structures of AtCPS, AgAS and ligand-binding state SmMDS, the residues H251 and C310 of OsCyc1 are located relatively distant from the substrate (Fig. 4f, 4g).” This is really interesting but including actual distances would be helpful to reader.
7. Line 288: “I311, L314, Y317 and H357 are at the bottom of GGPP in OsCyc1D367A structure.” Use more descriptive language since “bottom” is relative to viewpoint. Do you mean to say are near the aliphatic tail or C10-C15 of GGPP?
8. Line 313: “Besides, OsCyc15Mu can alter the stereochemistry of the product.” This sentence felt out of place—I had trouble finding the connection between the sentence before or after. Do you mean “in addition” instead of “besides”, or something to that effect?

Figures

9. Figure 3. Can you tell based on MW standards what molecular weight the shifted S674A/E677A peak is at? If so, I would label peaks.
10. Figure 4. “AG8 was shown only one conformation in the above pictures.” Clarify this sentence. I’m assuming that AG8 was modeled into that structure in multiple conformations which is what you are referring to—how much do the conformations differ? Are they of equal occupancy or is the conformation you’re showing more prominent? Also, I would just remind reader in legend that AG8 is the AtCPS substrate. Perhaps rephrase to something like “Although the substrate of AtCPS, AG8, was modeled in multiple conformations, only one conformation is shown in comparison figures.”
11. Figure 6. Make sure you use double-headed arrows instead of single-headed arrow in 6F mechanism! Single-head suggests those are single-electron transfers. Also I would change the bond

angle/length between C17 and the proton to be consistent with other angles.

Methods

12. Gene cloning: I'm assuming there is a His-tag based on the purification method—include that information in details about construct.
13. Include more details about protein expression: IPTG concentration, time and temperature of incubation after inducing overexpression.
14. Be sure to fully acknowledge not just Phenix, but the individual programs within the software package that you used for molecular replacement and refinement. For example, I am assuming you used Phenix Phaser (Line 435) for molecular replacement, and phenix.refine (Line 437) for computational refinement.
15. For molecular replacement, did you just use the complete AtCPS structure as search model or did you modify it at all (for example, prune side chains in Sculptor, remove waters/substrates/etc)? Were you searching for just a monomer? Give those details and programs used if so.
16. Also give more details on methods for computational refinement. What types of refinement were you doing—xyz reciprocal space, group or individual B-factors? Did you use NCS restraints in refinement since you have 6 molecules in asymmetric unit? Did your computational refinement method change as model improved?
17. Line 436: change to “manual model building” instead of “build”
18. Lines 485-489: Did you express and purify MDSD611A ad TwCPS3? If so, you need to include methods for expression and purification.

Overall I believe this is a strong paper on an interesting system, and I was excited to see the effort the authors put in to address and improve this manuscript. I recommend the paper to be accepted with minor revisions suggested above.

Reviewer #3 (Remarks to the Author):

This manuscript has been revised to address issues found in the original, but does not fully address these, and closer analysis, enabled by the somewhat better presentation, has raised a couple of others. For example, the abstract needs to clarify that class II diTPSs are not involved in all diterpenoid biosynthesis, only that of labdane-related diterpenoids. Perhaps more critically, it is incorrect that class II diTPSs do not require Mg²⁺. In fact, in all cases where this has been previously investigated Mg²⁺ has been found to be an important co-factor (see Peters et al, 2002, *Biochemistry*, 41:1836 & Prsic et al, 2007, *Plant Physiol.* 144:445). Thus, the description of similar investigation here needs to be modified accordingly (line 121). Similarly, stating that H275 is always a histidine is also incorrect. In fact, in most cases it is a leucine (as in AtCPS), such that H275 is found in just a couple of CPSs involved in more specialized metabolism in small grain cereals. Please correct – line 323. With regards to the revisions, these are not fully descriptive of the points they were meant to convey. For example, it is useful to know that the N-term truncation simply removes the plastid targeting peptide to alleviate any concerns about the effect on structure. Thus, on line 83 “which was” should be changed to “which essentially removes the plastid targeting peptide, yielding a pseudomature protein that is fully”. Mention of the hydrogen bond between D367 and N414 (line 213) should be preceded by description of the importance of this

interaction in appropriately positioning the carboxylic acid for protonation, as mentioned in ref. 23 and expanded upon in Koxsal et al (2014, *Biochim Biophys Acta*, 1840:184). Similarly, the distance between D367 and GGPP should be more precisely described to determine if this is the relevant interaction – i.e., between the d2-oxygen and carbon-14 (line 219). Also, the linear conformation of GGPP suggests this represents an early stage in binding, not the reaction per se (line 221). Moreover, the description of halimadienyl diphosphate production needs to be corrected. On line 341 “the carbon cation transfer” should be “rearrangement via 1,2-hydride and methyl shifts”. Finally, the authors should consider using the more prevalent “OsCPS4” rather than “OsCyc1”, which has fallen out of use, to refer to this enzyme (throughout the manuscript).

Reviewers' comments:

We sincerely appreciate the thorough review, valuable feedback, encouragement and recognition provided by the reviewers for our manuscript titled “Structural and functional investigations of *syn*-copalyl diphosphate synthase from *Oryza sativa*”. In the following sections, we will provide detailed responses to each comment and have duly incorporated the necessary revisions into the manuscript.

Reviewer #1 (Remarks to the Author):

The manuscript has been modified according to the reviewers' comments that were properly addressed.

Response: Thank you very much for recognizing our efforts in addressing the reviewer's comments. We sincerely appreciate the valuable advice and guidance you have provided throughout the entire process.

Reviewer #2 (Remarks to the Author):

In this manuscript, the authors present the first structures of CPPs that form *syn*-CPP and through biochemical and structural analysis, identifying residues that play a role in dictating stereochemistry of this product, focusing on OsCyc1. Furthermore, they explore how oligomeric state impacts activity, with OsCyc1 exhibiting a tetrameric state in solution vs other terpenoid synthases that are typically dimers or monomers in solution. Overall I think the authors have greatly strengthened this manuscript with supporting figures that show electron density (including composite omit maps from crystal structure) and clearly conveying both where strong electron density is observed and regions where side chain density is weaker and limited—these edits better allows the reader to know what conclusions can be drawn from these data. Furthermore, I appreciate the authors thorough responses to my major concerns. I am happy to recommend this manuscript for publication with minor revisions, clarification, and suggestions (see below):

Questions for clarification:

Comment 1. The authors added a cryo-EM structure of OsCyc1D367A in its hexameric form. For the wild-type structure, tetramers were the predominant species. Were hexamers the predominant oligomeric form for this variant, and if so, do the authors have any speculation as to what might have caused this difference? Were grids prepared the same way? Authors might want to comment on this aspect since they emphasize how OsCyc1 predominantly forms tetramers in solution.

Response: Thank you very much for your comments. During the protein purification process, we observed that the molecular weight of the elution peak of OsCyc1 in size exclusion chromatography corresponded to an oligomeric state, suggesting that OsCyc1's structure can be solved using cryo-EM method. Therefore, we attempted to elucidate the structures of both OsCyc1 and OsCyc1^{D367A} in complex with GGPP using cryo-EM. During the data processing,

we selected the largest particles after 2D classification, as we believed that other particles might represent incomplete complexes, ultimately resulting in the determination of the hexameric structures of OsCyc1 and OsCyc1^{D367A}.

Subsequently, by employing analytical ultracentrifugation analysis and static light scattering, we discovered that OsCyc1 exhibited distinct oligomeric states in solution, with the most prevalent oligomeric forms being trimeric or tetrameric states. This prompted us to reprocess the electron microscopy data of OsCyc1, and determined three structures of OsCyc1, namely dimer, tetramer and hexamer. However, we did not re-determine the structures of OsCyc1^{D367A}.

We greatly appreciate your valuable comment, as it prompted us to reevaluate the data of OsCyc1^{D367A}. We manually sorted the tetrameric and hexameric particles from 2D classification. Notably, there were more hexameric particles than tetrameric ones (with 999,716 particles for hexamers and 597,434 particles for tetramers). Ultimately, we solved the tetrameric structure of OsCyc1^{D367A} with C2 symmetry, utilizing a total of 221,146 particles, as presented in Table 1 and Figure 1 in this “Response to Reviewer” document. Thus, hexamers are the predominant oligomeric form for OsCyc1^{D367A} in cyro-EM particles.

Table 1: No. of particles that contribute to the map.

	dimer	tetramer	hexamer
OsCyc1	27984	149837	97919
OsCyc1 ^{D367A} +GGPP	-	221146	735747

Figure 1. Fourier shell correlation graphs and cryo-EM density maps used in model building. The maps are colored by local resolution.

Static light scattering and analytical ultracentrifugation analysis were performed using OsCyc1, and the results were consistent with the cryo-EM data. Therefore, we conclude that the major oligomeric state of OsCyc1 is tetrameric. To enhance precision, we have revised Lines 120-122 on Page 3 in the manuscript as follows:

“Combined with the cryo-EM structure results, these findings imply that the major oligomeric form of OsCyc1 in solution (as used in our experiments) is tetrameric.”

On the other hand, hexameric forms predominate in the OsCyc1^{D367A} cryo-EM sample. The grids were prepared in the same way for both OsCyc1 and OsCyc1^{D367A}, and we infer that grid preparation method is unrelated to the oligomer state. Furthermore, considering that the D367A mutation is located in inner of the active site, it may not impact oligomeric states.

Therefore, we speculate whether the substrate GGPP or its solution may have influenced the protein's oligomeric state. First, GGPP was stored in an ammonium salt buffer (MeOH: NH₄OH=70: 30) and added in excess (1.8 μL GGPP solution in a total protein solution volume of 20 μL). Therefore, we speculate that the addition of GGPP may have affected the pH and ion concentration of the solution, which could be the primary reason for the shift in oligomeric equilibrium towards hexamers. Second, hexamer might be the active state, while tetramer might be the inactive state. However, the mechanism by which GGPP and its solution influences protein oligomeric state remains unclear, making it an intriguing phenomenon deserving further investigation.

Comment 2. I was surprised by the experimental result that adding EDTA to OsCyc1 relative enzyme activity assays significantly inhibiting activity (Supplemental Fig 2 and 4b). If Mg²⁺ is proposed to partially inhibit relative activity, and highest activity is observed in the absence of additional Mg²⁺ ions, I would have hypothesized that adding EDTA to chelate additional Mg²⁺ would have restored some activity. I would recommend authors comment on this result of why EDTA strongly inhibits relative activity if Mg²⁺ is not essential.

Response: Thank you very much for your comments. Our results demonstrate that high concentrations of Mg²⁺ exert a partial inhibitory effect on the relative enzyme activity of OsCyc1. Furthermore, when magnesium ions were chelated by EDTA, there is a significant reduction in the relative enzyme activity. These experimental findings align with the impact of magnesium ions on other terpene synthases that reported.

For instance, one research (Prisic S, Peters RJ. *Synergistic substrate inhibition of ent-copalyl diphosphate synthase: a potential feed-forward inhibition mechanism limiting gibberellin metabolism. Plant Physiol. 2007 May;144(1):445-54.*) shows that the optimal Mg²⁺ concentration for AtCPS is approximately 1 mM (K_{cat}/K_m=2.6). Notably, the efficiency of AtCPS sharply decreases when 2 mM EDTA is introduced into the assay buffer (K_{cat}/K_m=0.001). Excessive Mg²⁺ significantly inhibits the enzyme activity of AtCPS (K_i=94±26 μM), while the inhibitory binding constant for AgAS was relatively lower (K_i=12000±4000 μM) (Figure 2 in this “Response to Reviewer” document.). This article suggests that the inhibitory effect of excessive Mg²⁺ on enzyme activity is likely mediated via “inhibitory Mg²⁺” binding to the DXDD motif. However, the mechanisms responsible for the reduced enzyme activity when Mg²⁺ are chelated were not discussed in the article.

Figure 2. Effect of magnesium on class II diterpene cyclase activity. A, Mg²⁺ dependence of rAtCPS activity. Inset depicts activity over the lower concentration range of Mg²⁺. B, Mg²⁺ dependence of rAgAS:D621A activity. (Prisic S, Peters RJ. Synergistic substrate inhibition of *ent-copalyl diphosphate synthase*: a potential feed-forward inhibition mechanism limiting gibberellin metabolism. *Plant Physiol.* 2007 May;144(1):445-54.)

Another article (Mann, Francis M. et al. A Single Residue Switch for Mg²⁺-dependent Inhibition Characterizes Plant Class II Diterpene Cyclases from Primary and Secondary Metabolism, *Journal of Biological Chemistry*, 2010, Volume 285, Issue 27, 20558 - 20563) cites two references (references 10 and 17 in the paper) and indicates that Mg²⁺ is essential for the enzyme to achieve full activity, with a suggested reason being “presumably for binding of the diphosphate moiety of their GGPP substrate”.

We infer that the impact of Mg²⁺ on OsCyc1 is more similar to that of AgAS than AtCPS. OsCyc1 requires low concentrations of Mg²⁺ for activity, and high concentrations partially inhibit the enzyme activity. We speculate that there are several possibilities that may lead to a decreased enzyme activity after chelating Mg²⁺ with EDTA for OsCyc1:

1. The pyrophosphate part of GGPP carries negative charges. It is worth noting that not all amino acids surrounding GGPP in the OsCyc1^{D367A} structure are positively charged, such as E199 (Figure 4e in the manuscript). We speculate that Mg²⁺ may bind to the pyrophosphate part of GGPP, aiding its correct positioning near E199 in the active site.
2. Another possibility is that Mg²⁺ may assist the substrate in entering the active pocket. As shown in Figure 3 in this “Response to Reviewer” document, there are two tunnels at the entrance to the active site. Tunnel 1 is predominantly negatively charged, which may not favor the pyrophosphate binding, while tunnel 2 is primarily positively charged, facilitating pyrophosphate binding. However, structural evidence shows that the pyrophosphate part of GGPP is positioned in tunnel 1. The specific mechanisms for these two tunnels remain unclear, but we infer that Mg²⁺ may assist GGPP in entering tunnel 1.

Figure 3. Electro potential map surface of the active site of OsCyc1^{D367A}. **a** Section view of active pockets. **b** Surface of the active pocket entrance. The red is for negative potential, white at zero, and blue is for positive in the electro potential map.

Comment 3. Lines 325-328: “These results indicated that H275 plays an important role in *syn*-CPP production. This speculation is in line with the comprehensive bioinformatics analysis of *syn*-CPSs, *ent*-CPSs and (+)-CPSs in plants, which shows that H275 is always histidine and different from other CPSs (Fig. 6e and Supplementary Fig. 21-23).” I was confused by the wording “H275 is always histidine”. It seems like you mean it is strictly conserved, but in the figures, it seems that despite being largely conserved, in some sequences, it is another residue (difficult to see in sequence logos what other residues are found at that position and whether they have similar chemical properties).

Response: Thank you very much for your comments. We sincerely apologize for any misunderstanding in the manuscript. Our intention is to convey that H275 is relatively conserved. As a result, we have revised the content on Page 8, Lines 338-341 accordingly:

“This speculation is in line with the comprehensive bioinformatics analysis of *syn*-CPSs, *ent*-CPSs and (+)-CPSs in plants, which shows that H275 is relatively conserved in *syn*-CPSs and different from other CPSs (Fig. 6e and Supplementary Fig. 21-23).”

In *syn*-CPSs, the H275 position is predominantly occupied by histidine; however, in a minority of enzymes, this position can also be substituted by phenylalanine, isoleucine, methionine, or tyrosine. These amino acids all share a common characteristic: the presence of a large side chain. Consequently, we propose that H275 plays a role in shaping the active site pocket and providing the requisite substrate-folding space for the synthesis of *syn*-CPP.

Comment 4. Lines 340-347: You mention how these residues including H501 are close to the C17 atom, and propose H501 to potentially act as a catalytic base. In the structure, the distance of H501 to C17 is 5 angstroms. Based on results and rationale, it does appear H501 probably plays an important role, but this distance seems far for a proton transfer—I wonder whether there is a closer potential base (another residue or perhaps even water acting as a base, activated by a nearby residue).

Response: Thank you for your insightful comments. In the docking results, we observed a relatively greater distance between H501 and C17, which would not favor deprotonation. This suggests the possibility of the presence of a nearby water molecule acting as catalytic bases. However, due to the limitations in resolution, it was not possible to construct the water molecule within the structure.

Our molecular docking was based on the structure of OsCyc1^{D367A}, in which the substrate is in a linear state. The active site conformation of OsCyc1 could change when the substrate folds. Hence, in the actual reaction, the distance between H501 and the substrate might differ from our docking results, and the possibility of it being closer cannot be ruled out.

Comment 5. The authors did put substantial effort into improving refinement of the structure. To address the gap between Rwork and Rfree, I am wondering if the authors have tried optimizing target X-ray/ADP or X-ray/stereochemistry weights during computational refinement? This can often help shrink that gap/reduce bias if model has become overrefined. I think it is better to have a slightly higher Rwork, but smaller gap between Rwork and Rfree, if possible to minimize bias.

Response: Thank you very much for your comments. We attempted to refine the X-ray/ADP weight or X-ray/stereochemistry weight during structure refinement. Using the X-ray stereochemistry weight successfully reduced the gap between Rwork and Rfree; however, this was accompanied by an increase in B-factors. To ensure the rationality of all parameters, we ultimately used the original 8KBW structure as the crystal structure.

Minor typos and suggestions:

Comment 1. When aligning structures of OsCyc1 with other CPPs and making statements of high similarity, including a quantitative RMSD would be helpful (for example, authors state in lines 98-100 that OsCyc1 is structurally similar to AtCPS, AgAS, and SmMDS—including RMSDs here or in the figure legend would be helpful).

Response: Thank you very much for your comments. We have included the RMSD values in the manuscript on Page 3, Line 101-104 as the following:

“The overall structure of monomer OsCyc1 is similar to that of AtCPS, AgAS and SmMDS, with RMSD values for C α of 1.86 Å, 2.15 Å and 1.94 Å, respectively (Supplementary Fig. 4a). The structure of monomer OsCyc1 consists of an $\alpha\beta\gamma$ domain and a catalytic center located between the β and γ domains (Fig. 1b).”

Comment 2. Lines 142-144: “The interactions between $\alpha 23$ and $\alpha 24$ include R652 with Q675, as well as E640 with K686, possibly through van der Waals interactions (Fig. 2c and Supplementary Fig. 6a-6c).” Would you expect those to only be van der Waals interactions or possibly a salt bridge between Glu and Lys?

Response: Thank you very much for your comments. The side chain densities of E640 and/or K686 are missing, making it impossible to determine their specific interactions. Since the

distance between residues involved in salt bridges is also considered critical, with the required distance being less than 4 Å, it's important to note that in the cryo-EM structure, both of these amino acids in chains A/B have distances greater than 4 Å. Therefore, we cannot confidently assert that these two residues are interacting through salt bridge interactions.

Comment 3. Lines 149-150, change “particles number” to “particle numbers”. In addition, I would refer the reader to supplemental table that shows particle number or even include in main text to fully convey this point.

Response: Thank you very much for your comments. We have made the correction on Page 4, Lines 154-156, changing “particles number” to “particle numbers”. Below each oligomer map in Figure 1c, the number of particles is indicated. Additionally, we have included Extended Data Table 3 in the Supplementary Information, which provides information on the number of particles contributing to the cryo-EM map. To provide the reader with a clearer understanding, we have included Figure 1c and Extended Data Table 3 at the end of this sentence, as follows (Page 4, Line 154-156):

“When we analyzed the particle numbers in the cryo-EM map reconstruction, we found that the tetramer is the predominant form of OsCyc1 and is equivalent to chains A/B/C/D in the crystal structure (Figure 1c and Extended Data Table 3).”

Comment 4. Lines 190-191: change to “showing that the tetramer seems to have a minor impact...”

Response: Thank you very much for your comments. We have made the change on Page 5, Line 197, as follows:

“showing that the tetramer seems to have a minor impact on enzyme activity *in vitro*.”

Comment 5. Lines 213-214: “In the structure of OsCyc1, K233, N405 and K453 interacts with the pyrophosphate, while M194 and E199 are close to the pyrophosphate of GGPP”. I was confused by this sentence—I would use more descriptive language for “close to” vs “interacts”. Are you saying that M194 and E199 might interact with the pyrophosphate—especially with E199 being a negatively charged group, would you expect it to interact with a negatively charged substrate moiety?

Response: Thank you very much for your comments. M194 is an amino acid with a hydrophobic side chain, which prevents it from forming strong interactions with the pyrophosphate. E199 is positioned 3.2 angstroms away from pyrophosphate, and both carry negative charges. We speculate that their interaction might occur through the involvement of other ions or water molecules. Therefore, in the text, we do not explicitly state an interaction between them but simply mention their close proximity.

Comment 6. Lines 242-244: “Remarkably, compared to the structures of AtCPS, AgAS and ligand-binding state SmMDS, the residues H251 and C310 of OsCyc1 are located relatively distant from the substrate (Fig. 4f, 4g).” This is really interesting but including actual distances

would be helpful to reader.

Response: Thank you very much for your feedback. We have included Extended Data Table 4 in the Supplementary Information and made the following change on Page 6, Line252-257:

“In OsCyc1, the side chains of H251, C310 and I311 are positioned at distances of 3.7 Å, 6.9 Å and 3.7 Å, respectively, from GGPP. In AtCPS, the corresponding amino acids have distances to AG8 of 2.6 Å, 4.6 Å and 4.2 Å, respectively, while in SmMDS, these distances are 3.5 Å, 6.3 Å and 3.6 Å, respectively (Extended Data Table 4). Therefore, compared to the structures of AtCPS and SmMDS, the residues H251, C310 and I311 of OsCyc1 are located relatively distant from the substrate (Fig. 4f, 4g).”

Comment 7. Line 288: “I311, L314, Y317 and H357 are at the bottom of GGPP in OsCyc1D367A structure.” Use more descriptive language since “bottom” is relative to viewpoint. Do you mean to say are near the aliphatic tail or C10-C15 of GGPP?

Response: Thank you very much for your comments. We have made the change on Page 7, Line302, as follows:

“I311, L314, Y317 and H357 are at the isoprene tail of GGPP in OsCyc1^{D367A} structure.”

Comment 8. Line 313: “Besides, OsCyc15Mu can alter the stereochemistry of the product.” This sentence felt out of place—I had trouble finding the connection between the sentence before or after. Do you mean “in addition” instead of “besides”, or something to that effect?

Response: Thank you very much for your comments. We have made the change on Page 8, Line326, as follows:

“Furthermore, OsCyc1^{5Mu} can generate not only *syn*-CPP but also *ent*-CPP.”

Figures

Comment 9. Figure 3. Can you tell based on MW standards what molecular weight the shifted S674A/E677A peak is at? If so, I would label peaks.

Response: Thank you very much for your comments. The standard elution profile of the SuperdexTM 200 HR 10/300 column is shown in Figure 4a in this “Response to Reviewer” document. The molecular weight of S674A/E677A is approximately 80.8 kDa. The elution peak for S674A/E677A appeared roughly between 14 mL and 15 mL, close to standard marker 3, which corresponds to 67 kDa. Based on these results, we inferred that S674A/E677A may be monomeric. However, due to slight variations in peak positions depending on the condition of the instrument and column, we couldn't directly determine its molecular weight from the SuperdexTM 200 HR 10/300 gel filtration alone, and the molecular weight is not indicated in Figure 3.

Figure 4. **a** Typical chromatogram from a function test of Superdex 200 10/300 GL. **b** Size exclusion chromatography of WT (OsCyc1⁶⁹⁻⁷⁶⁷) and S674A/E677A mutant.

Comment 10. Figure 4. “AG8 was shown only one conformation in the above pictures.” Clarify this sentence. I’m assuming that AG8 was modeled into that structure in multiple conformations which is what you are referring to—how much do the conformations differ? Are they of equal occupancy or is the conformation you’re showing more prominent? Also, I would just remind reader in legend that AG8 is the AtCPS substrate. Perhaps rephrase to something like “Although the substrate of AtCPS, AG8, was modeled in multiple conformations, only one conformation is shown in comparison figures.”

Response: Thank you very much for your comments. AG8 is an analog of GGPP. In the structure of AtCPS, AG8's pyrophosphate exhibits two conformations, oriented in different directions, both with an occupancy of 0.40. For clarity in the image, we display only one conformation. We have revised the legend of Figure 4 on Page 22, Line 829-830, as follows:

“In the structure of AtCPS, AG8’s pyrophosphate exhibits two conformations. For image clarity, we have opted to display only one conformation.”

Comment 11. Figure 6. Make sure you use double-headed arrows instead of single-headed arrow in 6F mechanism! Single-head suggests those are single-electron transfers. Also I would change the bond angle/length between C17 and the proton to be consistent with other angles.

Response: Thank you very much for your feedback. We apologize for the incorrect use of curved arrows. We have now replaced the arrows in Figure 6 with double-headed arrows. Additionally, we have adjusted the bond angle and length between C17 and the proton to ensure consistency with other angles.

Methods

Comment 12. Gene cloning: I’m assuming there is a His-tag based on the purification method—include that information in details about construct. Include more details about protein expression: IPTG concentration, time and temperature of incubation after inducing overexpression.

Response: Thank you very much for your comments. We have revised the manuscript on Page 10, Line 407-412 according to your comments:

“Codon-optimized genes encoding the full-length OsCyc1 protein and truncated protein (OsCyc1⁶⁹⁻⁷⁶⁷) were synthesized (GENEray biotechnology) and subcloned into pET-24 a (+) vectors between the *NdeI* and *XhoI* sites, including a His-tag. These constructs were subsequently transformed into BL21(DE3) cells and grown in LB medium at 37 °C until the optical density (OD)₆₀₀ reached 0.6-1.0. Protein expression was induced by addition of 0.2 mM isopropyl β-D-thiogalactoside (IPTG) for 18h-20h at 16 °C.”

Comment 13. Be sure to fully acknowledge not just Phenix, but the individual programs within the software package that you used for molecular replacement and refinement. For example, I am assuming you used Phenix Phaser (Line 435) for molecular replacement, and phenix.refine (Line 437) for computational refinement. For molecular replacement, did you just use the complete AtCPS structure as search model or did you modify it at all (for example, prune side chains in Sculptor, remove waters/substrates/etc)? Were you searching for just a monomer? Give those details and programs used if so. Also give more details on methods for computational refinement. What types of refinement were you doing—xyz reciprocal space, group or individual B-factors? Did you use NCS restraints in refinement since you have 6 molecules in asymmetric unit? Did your computational refinement method change as model improved? Line 436: change to “manual model building” instead of “build”

Response: Thank you very much for your comments. We have revised the manuscript on Page 11, Line 452-460 according to your comments:

“The structure was solved by molecular replacement using the structure of AtCPS (PDB ID: 3PYA, the model was modified by Chainsaw program in the CCP4 program suite⁴⁹) as a searching model by Phaser-MR (full-featured) in PHENIX⁵⁰ with the number of copies set to 4. Subsequently, Phaser-MR provided an interpretable electron density map, and the model was built using Phenix.Autobuild in PHENIX. The resulting structure was iteratively manually modeled and modified using Coot⁵¹ and refined by Phenix.Refine in PHENIX⁵⁰. The refinement strategy employed XYZ (reciprocal-space) and XYZ (real-space) refinement methods, along with Individual B-factors and Group B-factors. Additionally, non-crystallographic symmetry (NCS) was used. The strategy changed as the model improved. ”

Comment 14. Lines 485-489: Did you express and purify MDS611A and TwCPS3? If so, you need to include methods for expression and purification.

Response: Thank you very much for your comments. The expression and purification procedures of MDS^{D611A} and TwCPS3 are described on Page 10, Line 420-432 as follows:

“The D611A mutant of partially truncated miltiradiene synthase from *Selaginella moellendorffii* (SmMDS^{D611A}) and truncated kaurene synthase like from *Salvia miltiorrhiza* (SmKSL) were cloned into the pET-24 a (+) vectors between the *NdeI* and *XhoI* sites, including a His-tag. The construct was subsequently transformed into BL21(DE3) cells and grown in LB medium at 37 °C until the optical density (OD)₆₀₀ reached 0.6-1.0. Protein expression was induced by addition of 0.2 mM isopropyl β-D-thiogalactoside (IPTG) for 18h-20h at 16 °C.

SmMDS^{D611A} was purified by Ni affinity chromatography.

TwCPS3 from *Tripterygium wilfordii* (obtained from Gao's laboratory) was cloned into the pET-22b vector between the *NdeI* and *XhoI* restriction sites, including a His-tag. The construct was subsequently transformed into BL21(DE3) cells and grown in LB medium at 37 °C until the optical density (OD)₆₀₀ reached 0.6-1.0. Protein expression was induced by addition of 0.2 mM isopropyl β-D-thiogalactoside (IPTG) for 18h-20h at 16 °C. The recombinant protein was purified by Ni affinity chromatography and stored in buffer B.”

Comment 15. Overall I believe this is a strong paper on an interesting system, and I was excited to see the effort the authors put in to address and improve this manuscript. I recommend the paper to be accepted with minor revisions suggested above.

Response: Thank you for your thoughtful review and positive feedback. We greatly appreciate your recommendation for acceptance with minor revisions. Your constructive comments have been invaluable in improving our work.

Reviewer #3 (Remarks to the Author):

This manuscript has been revised to address issues found in the original, but does not fully address these, and closer analysis, enabled by the somewhat better presentation, has raised a couple of others.

Comment 1. For example, the abstract needs to clarify that class II diTPSs are not involved in all diterpenoid biosynthesis, only that of labdane-related diterpenoids.

Response: Thank you very much for your comments. To maintain consistency in the abstract and incorporate the points you mentioned, we have included a description in the introduction section, clarifying that class II diTPSs are not involved in all diterpenoid biosynthesis, as follows, on Page 1-2, Lines 39-43:

“Diterpene synthases can be categorized into two classes, with class I diterpene synthases being ion-initiated, while class II diterpene synthases are proton-initiated. Although not all diterpene biosynthesis processes involve class II diterpene synthases, most medicinal plants produce diterpenes with a labdane skeleton, and their synthesis involves the catalysis of class II diterpene synthases⁴.”

Comment 2. Perhaps more critically, it is incorrect that class II diTPSs do not require Mg²⁺. In fact, in all cases where this has been previously investigated Mg²⁺ has been found to be an important co-factor (see Peters et al, 2002, *Biochemistry*, 41:1836 & Prsic et al, 2007, *Plant Physiol.* 144:445). Thus, the description of similar investigation here needs to be modified accordingly (line 121).

Response: Thank you very much for your comments. As we responded to Reviewer #2's Comment 2, diterpene synthases require trace amounts of magnesium ions. We have made changes to the content on Pages 3, Lines 127-129, based on your suggestions, and we have also

added new literature to the references. The revised content is as follows:

“Considering that Mg^{2+} is essential for class I terpene synthases but generally not required in excess for most class II terpene synthases^{27, 28}, we assessed the relative enzyme activity of OsCyc1 at different Mg^{2+} concentrations.”

Comment 3. Similarly, stating that H275 is always a histidine is also incorrect. In fact, in most cases it is a leucine (as in AtCPS), such that H275 is found in just a couple of CPSs involved in more specialized metabolism in small grain cereals. Please correct – line 323.

Response: Thank you very much for your comments. We apologize for any confusion caused by the unclear expression in our article. What we meant is that H275 is always histidine in *syn*-CPSs. In accordance with your and Reviewer #2’s suggestions, we have revised Lines 339-341 on Page 8 as follows:

“This speculation is in line with the comprehensive bioinformatics analysis of *syn*-CPSs, *ent*-CPSs and (+)-CPSs in plants, which shows that H275 is relatively conserved in *syn*-CPSs and different from other CPSs (Fig. 6e and Supplementary Fig. 21-23).”

Comment 4. With regards to the revisions, these are not fully descriptive of the points they were meant to convey. For example, it is useful to know that the N-term truncation simply removes the plastid targeting peptide to alleviate any concerns about the effect on structure. Thus, on line 83 “which was” should be changed to “which essentially removes the plastid targeting peptide, yielding a pseudomature protein that is fully”.

Response: Thank you very much for your comments. We removed the N-terminal sequence based on the prediction of the protein’s secondary structure, eliminating irregular disordered regions to enhance protein stability. This process also eliminated the plastid targeting peptide. As a result, following your advice, we made revisions to Lines 86-89:

“To determine the structure of OsCyc1, we used the stable N-terminal truncation variant of OsCyc1, OsCyc1⁶⁹⁻⁷⁶⁷, from which the plastid targeting peptide was removed. This resulted in a pseudo-mature protein that was catalytically active as shown by our relative enzyme activity assay (Supplementary Fig. 1 and 2).”

Comment 4. Mention of the hydrogen bond between D367 and N414 (line 213) should be preceded by description of the importance of this interaction in appropriately positioning the carboxylic acid for protonation, as mentioned in ref. 23 and expanded upon in Koksal et al (2014, Biochim Biophys Acta, 1840:184).

Response: Thank you very much for your comments. We have made the corresponding revisions based on your suggestions and have included new references. The specific changes can be found on Page 6, Lines 220-224, as follows:

“In terpene synthases, the middle aspartic acid of the DXDD motif is typically stabilized by

nearby basic amino acids. For instance, in squalene-hopene cyclase (SHC) from *Alicyclobacillus acidocaldarius*²⁹ and bacterial diterpene synthases such as PtmT2³⁰, the basic amino acid is histidine, whereas in plant class II terpene synthases, it is asparagine^{31, 32} (Supplementary Fig. 15).”

Comment 4. Similarly, the distance between D367 and GGPP should be more precisely described to determine if this is the relevant interaction – i.e., between the d2-oxygen and carbon-14 (line 219). Also, the linear conformation of GGPP suggests this represents an early stage in binding, not the reaction per se (line 221).

Response: Thank you very much for your constructive feedback. Based on your suggestions, we have made revisions to the following content (Page 6, Lines 224-235).

“In OsCyc1^{D367A} structure, the DXDD motif and the conserved basic amino acid, asparagine (N414), are situated at the terminus of the isoprene tail of GGPP. The distance between the d2-oxygen atom in D365 and the carbon-20 atom in the isoprene tail of GGPP is 2.59 Å (Supplementary Fig. 14d). However, measuring the distance between the unmutated D367 and the substrate is not feasible due to its mutation to alanine. In the crystal structure of OsCyc1, the distance between the d1-oxygen in D367 and the d2-nitrogen in the conserved N414 is 2.97 Å (Supplementary Fig. 14e, 14f). Upon superimposing the crystal structure and the OsCyc1^{D367A} structure, the measured distance between the d2-oxygen in D367A and the carbon-19 in GGPP is 2.82 Å, which is suitable for initiating the cyclization reaction (Supplementary Fig. 14g). However, in the OsCyc1^{D367A} structure, GGPP adopts a linear conformation, possibly reflecting the stage of binding, which might not represent the cyclization state.”

Comment 5. Moreover, the description of halimadienyl diphosphate production needs to be corrected.

Response: Thank you very much for your comments. We have replaced “halimadienyl diphosphate” with “halima-5,13*E*-dienyl diphosphate” in the revised manuscript and Supplementary Figures 15 and 20.

Comment 6. On line 341 “the carbon cation transfer” should be “rearrangement via 1,2-hydride and methyl shifts”.

Response: Thank you very much for your comments. We have revised Lines 357-359 on Page 9 as follows:

“*Syn*-halima-5,13*E*-dienyl diphosphate is the product quenched from the intermediate halima-13*E*-en-10-yl⁺, which is derived from the rearrangement via 1,2-hydrid and methyl shift of the intermediate lambda-13*E*-en-8-yl⁺⁴³ (Supplementary Figure 20c).”

Comment 7. Finally, the authors should consider using the more prevalent “OsCPS4” rather than “OsCyc1”, which has fallen out of use, to refer to this enzyme (throughout the manuscript).

Response: Thank you very much for your comments. We will emphasize that OsCPS4 is

commonly used in the revised paper, as seen in Lines 47-48. Additionally, considering that the PDB and EMDDB data associated with the article use the name “OsCyc1”, we will continue to use the name OsCyc1 in this article and we plan to update the protein description to OsCPS4 in future research.

The content of Lines 47-48 is as follows:

“In the common cereal rice (*Oryza sativa*), *syn*-copalyl diphosphate synthase (also commonly known as OsCPS4 or OsCyc1) catalyzes GGPP to *syn*-CPP⁷⁻⁹ (Fig. 1a).”

REVIEWERS' COMMENTS:

Reviewer #2 (Remarks to the Author):

The authors have addressed all questions and concerns thoroughly, and I believe these additions and changes greatly strengthen the paper. I am happy to recommend this paper be accepted for publication.

Reviewer #3 (Remarks to the Author):

This manuscript has been re-revised to address issues found in the original, but does not fully address these. In particular, it is still necessary for the abstract to clarify that class II diTPSs are not involved in all diterpenoid biosynthesis, only that of labdane-related diterpenoids, which could be done by simply replacing “An essential step in diterpene biosynthesis” with “The large superfamily of labdane-related diterpenoids are defined by” (line 19). The added description of the role of Mg²⁺ in class II diTPSs also should be clarified, as this serves as a co-substrate, likely two with every substrate (see Pan et al, 2022, JACS 144:22067), which should be specified here (line 128).

Response to Reviewers

We greatly appreciate the recognition of our work from the editors and reviewers. We sincerely thank the editors and reviewers for your comprehensive review and valuable feedback on our manuscript titled “Structural and functional investigations of *syn*-copalyl diphosphate synthase from *Oryza sativa*”. Your expertise and invaluable suggestions have significantly enhanced the quality of our article. Below, we provide a point-by-point response to all of the comments.

Reviewer #2 (Remarks to the Author):

Comment The authors have addressed all questions and concerns thoroughly, and I believe these additions and changes greatly strengthen the paper. I am happy to recommend this paper be accepted for publication.

Response: Thank you for your comprehensive review and invaluable feedback. We greatly appreciate your recommendation for acceptance.

Reviewer #3 (Remarks to the Author):

Comment 1 This manuscript has been re-revised to address issues found in the original, but does not fully address these. In particular, it is still necessary for the abstract to clarify that class II diTPSs are not involved in all diterpenoid biosynthesis, only that of labdane-related diterpenoids, which could be done by simply replacing “An essential step in diterpene biosynthesis” with “The large superfamily of labdane-related diterpenoids are defined by” (line 19).

Response: Thank you very much for your comments. We have made the change in the abstract, replacing “An essential step in diterpene biosynthesis” with “The large superfamily of labdane-related diterpenoids is defined by”. The revised content is as follows:

“The large superfamily of labdane-related diterpenoids is defined by the cyclization of linear geranylgeranyl pyrophosphate (GGPP), catalyzed by copalyl diphosphate synthases (CPSs) to form the basic decalin core, the copalyl diphosphates (CPPs).”

Comment 2 The added description of the role of Mg²⁺ in class II diTPSs also should be clarified, as this serves as a co-substrate, likely two with every substrate (see Pan et al, 2022, JACS 144:22067), which should be specified here (line 128).

Response: Thank you very much for your comments. We have integrated your recommendations and included the reference you mentioned. The updated content is as follows:

“Initially, we assessed the relative enzyme activity of OsCyc1 at different concentrations of Mg²⁺. According to reports, Mg²⁺ is essential for class I terpene synthases, while for class II

terpene synthases, an excessive amount of Mg^{2+} is generally not required, and trace amounts of Mg^{2+} are necessary^{27, 28}. Moreover, it was reported that in class II sesquiterpene cyclase, magnesium ions function as co-factors, typically two with every substrate²⁹. ”